# Estimating the variability of deep ocean particle flux collected by sediment traps using satellite data and machine learning

Théo Picard [1], Chelsey A. Baker [2], Jonathan Gula [3,4], Ronan Fablet [5], Laurent Mémery [1], and Richard Lampitt [2]

[1]Univ Brest, CNRS, IRD, Ifremer, Laboratoire des Sciences de l'Environnement Marin (LEMAR), IUEM, Plouzané, France
[2]National Oceanography Centre, Southampton, United Kingdom
[3]Univ Brest, CNRS, IRD, Ifremer, Laboratoire d'Océanographie Physique et Spatiale (LOPS), IUEM, Plouzané, France
[4]Institut Universitaire de France (IUF), Paris, France
[5]IMT Atlantique, Lab-STICC, Plouzané, France

**Correspondence:** Théo Picard (theo.picard@univ-brest.fr)

**Abstract.**

The gravitational pump plays a key role in the ocean carbon cycle by exporting sinking organic carbon from the surface to the deep ocean. Deep sediment trap time-series provide unique measurements of this sequestered carbon flux. Sinking particles are influenced by physical short-term spatio-temporal variability, which inhibits the establishment of a direct link to their surface origin. In this study, we present a novel machine learning tool, designated as $Unet_{sst-ssh}$, which is capable of predicting the catchment area of particles captured by sediment traps moored at a depth of 3000 m above the Porcupine Abyssal Plain (PAP), based solely on surface data. The machine learning tool was trained and evaluated using Lagrangian experiments in a realistic CROCO numerical simulation. The conventional approach of assuming a static 100-200 km box over the sediment trap location, only yields an average prediction of $\sim 25\%$ of the source region, whilst $Unet_{sst-ssh}$ predicts $\sim 50\%$. $Unet_{sst-ssh}$ was then applied to satellite observations to create a 20-year catchment area dataset, which demonstrates a stronger correlation between the PAP site deep particle fluxes and surface chlorophyll-a concentration, compared with the conventional approach. However, predictions remain highly sensitive to the local deep dynamics which are not observed in surface ocean dynamics. The improved identification of the particle source region for deep ocean sediment traps can facilitate a more comprehensive understanding of the mechanisms driving the export of particles from the surface to the deep ocean, a key component of the biological carbon pump.

## 1 Introduction

The biological carbon pump (BCP) is one mechanism that sequesters carbon from the atmosphere into the deep ocean. The BCP plays a key role in the climate system, as without it the atmospheric $CO_2$ concentrations would be about twice those observed today (Parekh et al., 2006; Kwon et al., 2009). Furthermore, the BCP is a crucial source of food resources in the deep ocean (Grabowski et al., 2019). However, despite the considerable importance of the BCP, the driving mechanisms are poorly understood (Le Moigne, 2019). Given that climate change driven perturbations may have wide-scale implications for the BCP,

it is of utmost importance to improve our understanding of this topic (Kwon et al., 2009; Passow and Carlson, 2012; Palevsky and Nicholson, 2018; Henson et al., 2022; Wilson et al., 2022).

One of the main processes contributing to the export of the BCP is the export of organic particles from the surface to the deep ocean, which sink due to their excess density (Siegel et al., 2016; Durkin et al., 2016; Le Moigne, 2019). This is known as the gravitational pump (Boyd et al., 2019; Siegel et al., 2023). This is a complex process modulated on one hand by phytoplankton net primary production (NPP), which uses carbon dioxide, solar energy and available nutrients for photosynthesis in the lighted upper layer of the ocean, also known as the euphotic zone ($\sim$0–200 m), and on the other hand by zooplankton faecal pellets (Lampitt et al., 1990). To assess the magnitude and composition of particles sinking via gravitational pump, long-term observations of the downward particle flux have been made using moored sediment traps (STs). These have been widely used to measure deep particle fluxes below 2000 m (Honjo et al., 2008; McDonnell et al., 2015). At this depth, the carbon can be sequestered for decades or centuries (Guidi et al., 2015; Burd et al., 2016; Siegel et al., 2021; Baker et al., 2022). However, while the time-series data from the STs are crucial for estimating the amount of long-term carbon sequestration and understanding the evolution of the global carbon cycle, fluxes from STs are often generalised over a wide spatial area despite being only a single data location. This spatial limitation hinders the ability of these instruments to capture the inherent variability of deep ocean particle fluxes. Indeed, medium and small local dynamics affect the sinking particles pathways and can have a significant impact on the ST measurements, especially over short time periods (Siegel et al., 1990; Deuser et al., 1990; Burd et al., 2010; Liu et al., 2018; Dever et al., 2021; Wang et al., 2022a). This means that particles originate over a large area of the surface ocean, called the catchment area (Deuser et al., 1988; Waniek et al., 2000), highly dependent on the local currents throughout the water column. It therefore remains a challenge to establish a clear link between observed NPP at the surface and deep carbon fluxes (Lampitt et al., 2010, 2023). This is particularly true for 10–30 day time periods, during which the drivers of carbon "pulses" observed in the STs remain unexplained (Smith et al., 2018).

This study focuses on the contribution of the local physics to the gravitational sinking flux. Traditionally, the sinking particles catchment area is typically represented as a 100 or 200 km box around the ST (Armstrong et al., 2001; Lampitt et al., 2010, 2023). This simplified catchment area is based on several studies that have used Lagrangian particle backtracking with physical model fields over several years (Waniek et al., 2000; Siegel et al., 2008; Wekerle et al., 2018; Wang et al., 2022a) to define a so-called "statistical funnel". The statistical funnel may allow for the annual surface area that influences sediment trap measurements to be captured but it does not capture the mesoscale spatial variability with timescales of weeks to months. So far, the only method capable of capturing this variability is that of Lagrangian backtracking experiments in reanalyses, i.e. the release of Lagrangian particles in a numerical simulation forced with observations that are supposed to represent the full 3D dynamics of the ocean (Frigstad et al., 2015; Liu et al., 2018; Ruhl et al., 2020; Ma et al., 2021). However, the practice of Lagrangian backtracking in reanalyses has a number of caveats:

– Reconstruction of mesoscale and submesoscale sea surface dynamics in numerical models, especially below 150 km resolution, remains a challenge for operational systems with data assimilation schemes (Lellouche et al., 2021; Cutolo et al., 2022; Febvre et al., 2023), which can lead to significant biases in the Lagrangian transport, usually unquantified.

- The deep dynamics (below 1000 m) is typically not validated, due to lack of observational data and/or understanding, and is almost completely absent in some data assimilation models (Lellouche et al., 2021). Our understanding of the influence of this phenomenon and how well it is represented in models is very limited.

- The process of reanalysis is typically complicated and computationally demanding, especially when used in conjunction with backtracking Lagrangian studies. This inherent complexity leads to certain constraints, such as the use of only a single particle sinking velocity or a limited time frame for the experiments.

To address the aforementioned problems, we have developed a new tool based on machine learning to predict the catchment area of particles reaching deep ocean STs, directly from the model output surface data (Picard et al., 2024). This approach was motivated by two main advances from the literature. Firstly, Wang et al. (2022a) showed that the monthly catchment area is closely related to the surface mesoscale dynamics, and in particular to local eddies observed with satellite altimetry (Chelton et al., 2011). In addition, recent studies have demonstrated the benefits of machine learning in predicting ocean interior currents from surface observations (Chapman and Charantonis, 2017; Bolton and Zanna, 2019; Manucharyan et al., 2021), as well as its high performance in reconstructing Lagrangian particle trajectories (Jenkins et al., 2022). Picard et al. (2024) trained a neural network with a numerical simulation dataset at the Porcupine Abyssal Plain sustained observatory (PAP-SO) Station, situated in the Northeast Atlantic Ocean (49 N, 16.5 W). The PAP-SO site has collected more than 30 years of deep ocean particulate organic carbon flux time series (Hartman et al., 2021; Lampitt et al., 2023). Picard et al. (2024) demonstrated the ability to predict the catchment area for particles with a sinking rate of w=50 m.d$^{-1}$ collected in a PAP-SO ST at 1000 m, using only surface numerical simulation outputs. Furthermore, a framework was presented to evaluate the prediction efficiency depending on the local physical conditions, with the best predictions associated with low kinetic energy and the presence of mesoscale eddies above the ST.

Therefore, this study has two main objectives. The first one is to improve the methodology presented in Picard et al. (2024), by proposing an enhanced version of the machine learning model that is capable of predicting the catchment area of particles collected at 3000 m by the PAP-SO station ST, taking into account a wider range of particle sinking velocities (Section 2). Indeed, as previously stated by Wekerle et al. (2018), the provenance of particles can vary considerably depending on their sinking velocity. Consequently, it is imperative to consider the entire particle velocity spectrum in order to accurately represent all the possible source areas. We also chose to focus on a 3000 m ST because the PAP-SO deep particle flux dataset is the most complete. Indeed, STs at 1000 m at the PAP-SO station do not give reliable results likely due to hydrodynamic biases for conical traps in the upper ocean (Buesseler et al., 2007) whilst fluxes collected at 3000 m are much more reliable. Similarly to Picard et al. (2024), we will evaluate the network performance and identify the physical factors that influence the accuracy of the catchment area prediction (Section 3). Considering that the dynamics below 1000 m at PAP-SO is weak compared to the upper layer (Wang et al., 2022a), we expect similar results to Picard et al. (2024), with the particle sinking velocity as the primary factor influencing the prediction score. The second objective is to investigate whether the connection between satellite-derived surface chlorophyll-a concentration, as a proxy for phytoplankton biomass, and the deep ocean ST fluxes can be improved with the application of the trained machine learning tool (Section 4).

## 2   Methods

In this study, we follow the methodology presented in Picard et al. (2024), where we use a series of Lagrangian experiments in a numerical simulation at the PAP-SO station to train Convolutional Neural Networks (CNNs) to predict the origin of particles collected in a deep ocean ST. We have adapted the learning strategy to train a model that can be applied to satellite data. This section presents the experiments carried out and the characteristics of the CNNs used.

### 2.1   Numerical simulation and Lagrangian experiments

The North Atlantic Subpolar Gyre simulation (POLGYR), designed and validated by Le Corre et al. (2020), is used in this study. This simulation is run using the Coastal and Regional Ocean COmmunity (CROCO) model, based on the Regional Ocean Modeling System (ROMS) (Shchepetkin and McWilliams, 2005). The grid has a horizontal resolution of 2 km and 80 vertical levels, allowing the simulation to fully resolve the mesoscale processes and partly resolve of the submesoscale. The focus of this study is the PAP-SO, represented by the black 1020 km square centred at the PAP-SO station (49°N, 16.5 °W) (see Figure 1a).

A series of Lagrangian backtracking experiments were performed to represent the sinking particle pathways from the surface ocean to the PAP-SO sediment trap at 3000 m. In order to account for the wide range of particle sinking velocities observed in the region, as reported in Villa-Alfageme et al. (2016), the experiments were performed with five different sinking velocities $w$, namely 80, 100, 150, 200 and 300 m.d$^{-1}$. Although slower sinking particles (w < 80 m.d$^{-1}$) have been observed at PAP-SO (Baker et al., 2017; Villa-Alfageme et al., 2016), they are not considered in this study due to computational constraints. Slower sinking particles present a significant challenge in terms of time taken to sink to 3000 m and dispersion in the spatial dimension, which in turn increases the size of our model domain and output considerably.

The Lagrangian experiment is carried out according to the general methodology presented in Picard et al. (2024) considering here a deeper ST depth and several particle sinking velocities. Over a period of 10 days, representing the ST collection period, 720 particles (36 particles every 12 hours) are released at the PAP-SO sediment trap, which is moored at a depth of 3000 m. During the experiment, all particles have a constant sinking velocity $w$. Once the particles have ascended to a depth of 200 m, which defines the depth of effective particle export (Wang et al., 2022a), their position is recorded (Figure 1b) and the probability density function (PDF) associated with this position is computed. The PDF represents the catchment area of the particles captured by the sediment trap during the 10-day collection period. This is also the variable predicted by the convolutional neural networks (CNNs). For each $w$ considered in this study, a total of 10,260 independent Lagrangian experiments were performed, each providing a PDF associated with a different dynamical condition. Further details on the methodology used can be found in (Picard et al., 2024).

### 2.2   Convolutional Neural Network architecture and training scheme

We have trained different CNNs to predict the catchment area, depending on the sinking velocities ($w$) considered here. The training methodology follows a state-of-the-art scheme with independent training, validation and test datasets (Lecun et al.,

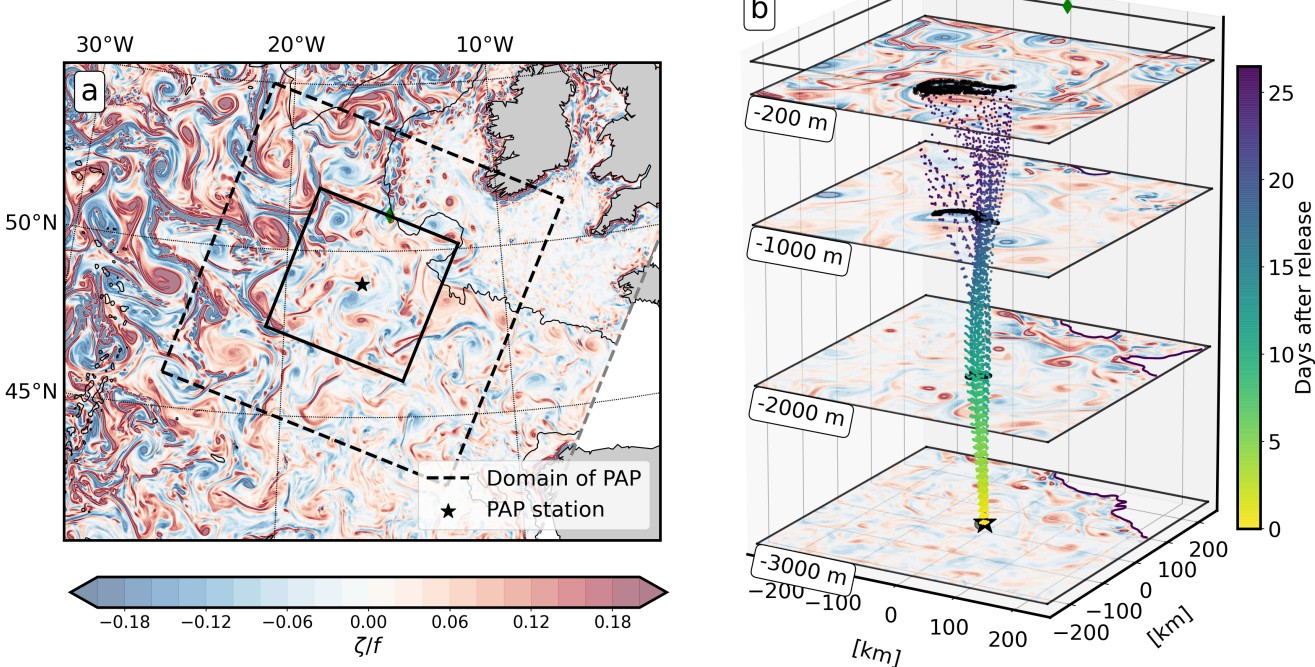

**Figure 1.** (a) Surface snapshot of relative vorticity in the numerical simulation. The black star represents the location of the PAP-SO station. The dashed square outlines the domain considered in this study. (b) A closer examination of the solid black square, with a focus on the vertical dimension. Relative vorticity at 200 m, 1000 m, 2000 m and 3000 m depth. The location of the sediment trap is indicated by the black star. A group of particles from a single Lagrangian experiment is shown. The colours of the particles represent the time in days after the release at the ST trap. When the particles reach a depth of 200 m, their position is saved (black dots) to compute the two-dimensional probability density function (PDF). The green diamond indicates the north-east of the sub-domain, with the PAP station location as the reference point.

2015). We used U-net schemes as described in Ronneberger et al. (2015). These schemes are among the state-of-the-art neural architectures for mapping problems with n-dimensional tensors, with numerous applications in imaging science (Falk et al., 2019), as well as recent applications in ocean science (Lguensat et al., 2018; Beauchamp et al., 2022; Jenkins et al., 2022). For 125 each training run, we use 8604 Lagrangian experiments for training, 1224 for validation, and 6800 for testing. Further details of the methodology can be found in Picard et al. (2024). To evaluate our predictions, we consider the Bhattacharyya coefficient (Bhattacharyya, 1943) to assess the similarity between the true pdf and the predicted one:

$$BC_z = \Sigma^{i \in D} \sqrt{P_{i,z} Q_{i,z}} \tag{1}$$

where $D$ represents the PAP domain, $P_i$ is the predicted PDF value and $Q_i$ is the true PDF computed from the Lagrangian 130 experiment at point i and at depth z. The Bhattacharyya coefficient is used to evaluate the similarity between two PDFs and serves as the loss function. In the following section, we refer to this loss function as the Bhattacharyya training loss (BL).

$$BL_{200m} = 1 - BC_{200m} = 1 - \Sigma\sqrt{P_{i,200m}Q_{i,200m}} \tag{2}$$

$BL_{200m}$ ranges from 1 to 0, with 0 representing a perfect prediction. We implement our machine learning scheme using PyTorch (Paszke et al., 2019). The training phase relies on Adam optimizer (Kingma and Ba, 2015) with the following hyperpa-
135 rameters: $\beta = (0.5, 0.999)$, no weight decay, and a learning rate of 0.001. The training process is performed using mini-batches of size 32. After 50 training epochs, the best model is selected based on its performance on the validation dataset. We further improve the performance and robustness of the model by using a bootstrapping method with 10 replicates (Breiman, 1996). The final prediction is a set of PDFs computed as the median of the predictions from the 10 models, followed by a re-normalization step.

The inputs of the Unets are geophysical fields for a 800 km-wide square box around the sediment trap with a 50-day time window and a 10-day time step. Three different Unet models were used to evaluate the impact of the input type and resolution:

- $Unet^w_{5V-4L}$: this configuration uses 5 variables as inputs, namely temperature, sea surface height (SSH), horizontal velocities U and V, and vorticity at a horizontal resolution of 8 km and at 4 vertical levels (excepted for SSH) (0 m, 750 m, 1500 m, 2250 m).

- $Unet^w_{5V-1L}$: this configuration uses sea surface only fields as inputs, namely SST, SSH and sea surface velocities at a horizontal resolution of 8 km.

- $Unet^w_{sst-ssh}$: this configuration uses only SST and SSH as inputs. Its training involves spatially-averaged fields to account for the effective resolution of satellite-derived products in the region (80 km for SSH (Chelton et al., 2011); 28 km for SST L4 product.

Of these three models, we expect $Unet^w_{sst-ssh}$ to be more applicable to reanalysis and satellite-derived products as it has been trained under conditions consistent with observational data inputs. The other two models will allow us to explore the key drivers of Lagrangian particle trajectories from the surface to the deep ocean. In addition to these Unet models, prediction baselines are considered in the form of 100 km and 200 km boxes centred at the PAP-SO station, denoted $box_{100km}$ and $box_{200km}$ (i.e. a PDF with uniform values within the box). These baselines represent the conventional approach that has
traditionally been used in previous studies to represent the particles' surface origins (Frigstad et al., 2015; Lampitt et al., 2023) and are used here as a reference point to assess the added value of the CNNs.

### 2.3 Test dataset and evaluation metrics

The considered test dataset consists of 6800 independent Lagrangian experiments that are used for testing the CNNs. Based on the $BL_{200m}$ score introduced in (Picard et al., 2024), we define a binary classification score as an evaluation metric:

- if $BL_{200m} < 0.3$ : the prediction is valid

– if $BL_{200m} \geq 0.3$ : the prediction is non-valid

As shown in (Picard et al., 2024), the $BL_{200m}$ score is directly linked to the overlap between the two distributions defined as :

$$F_{200m} = \Sigma^{i \in D} min(P_{i,200m}, Q_{i,200m}) \qquad (3)$$

The criterion of $BL_{200m} < 0.3$ is arbitrarily chosen to represent a valuable prediction, such that the prediction accounts for $F_{200m} = 45\%$ or more of the particles. The prediction made by $Unet^w_{sst-ssh}$ for the simulation test dataset will be referred to as $D^w_{simu}$ in the following. Figure 2 shows three samples from this dataset. The predictions are compared with the PDF of the true particle origins from the Lagrangian experiments (see Section 2.1). In this example, predictions (a) and (b) are considered valid, while prediction (c) is considered invalid.

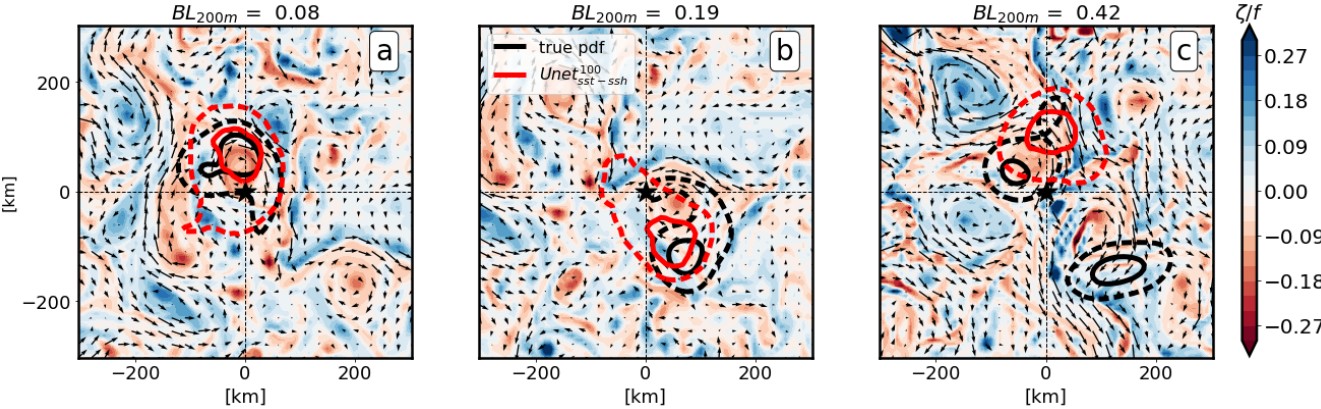

**Figure 2.** Examples of predictions of the probability density function (PDF) of particle origins from the $D^w_{simu}$ simulation-based dataset. The PDFs are represented by two contours: the solid contour represents 25% of the integrated PDF, while the dashed contour represents 75%. The black PDFs are the true PDFs derived from the Lagrangian experiment, and the red PDFs are the predictions using $Unet^{100}_{sst-ssh}$. We report the corresponding Bhattacharyya scores. The background represents the relative vorticity 20 days after the initial particle release, which coincides with the particles reaching the euphotic layer (z = 200 m) with a sinking velocity of 100 m.d$^{-1}$. Be advised that in scenario c), the true PDF is split into two patches. This is likely due to divergent dynamics at the source point located at the junction of several eddies, which makes the prediction more challenging.

## 170    3    Sensitivity analysis on simulation datasets

In this section, we evaluate the performance of the different Unet schemes. We test the robustness of the predictions while varying the horizontal resolution of the inputs, the particle sinking velocity and the type of inputs. Our aim is to gain a deeper understanding on the key influences on sinking particle trajectories.

## 3.1 Impact of input drivers and associated spatial resolutions

We first focus on a sinking velocity of w = 100 m.d$^{-1}$, which has been assumed to be the mean velocity of particles sinking to the deep ocean observed at the PAP-SO station (Lampitt et al., 2001; Villa-Alfageme et al., 2014, 2016). To evaluate the robustness of the predictions with respect to the horizontal resolution of the input variables, we examine the evolution of the prediction score given by $Unet_{sst-ssh}^{100}$ (Figure 3) by progressively degrading the effective resolution of the inputs SST (black dashed line) and SSH (red dashed line) fields from 8 km (effective resolution of the numerical simulation) to 200 km. The downscaling is conducted using an under-sampling method. To isolate the impact for each dataset, the SST resolution is fixed at 24 km when the SSH resolution is downscaled, and vice versa when the SST resolution is degraded, the SSH resolution is fixed at 80 km. The evaluation is performed by computing the percentage of valid predictions from the entire test dataset. The score does not change significantly with SST resolution, whereas the score decreases significantly with coarser SSH resolution. We conclude that the information from SSH, which includes geostrophic currents information, is the main driver for particle trajectory predictions. Conversely, the information derived from SST, which provides smaller scale features such as fronts, seems to play a secondary role. Regarding the resolution of SSH, the prediction score is not significantly affected at a resolution of 80 km compared to a finer resolution (a loss of about 3% of valid predictions). This result supports the potential application of the trained models with real satellite-derived products. However, it is important to note that for SSH resolutions greater than 100 km, the network prediction score can be seriously degraded.

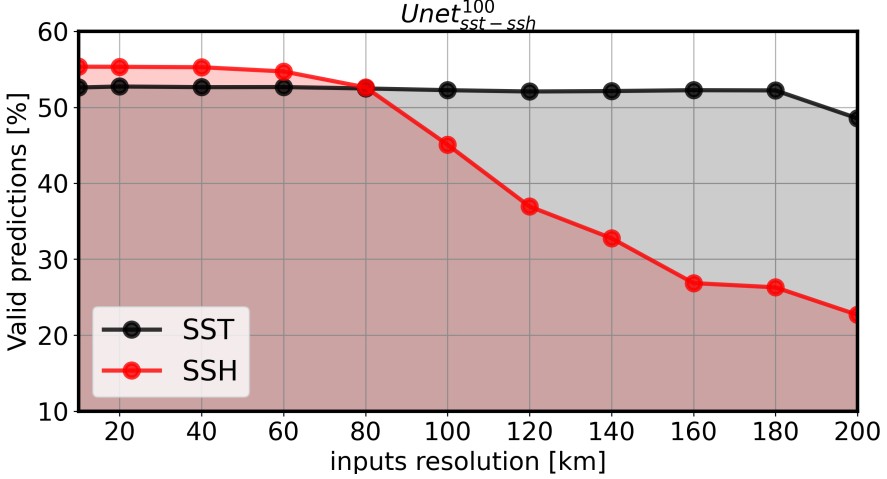

**Figure 3.** Evaluation of $Unet_{sst-ssh}^{100}$ score as a percentage of the valid predictions computed with the numerical simulation test dataset. The axis represents the horizontal resolution of the inputs downscaled using an under-sampling method. When the SSH resolution is downscaled, the SST resolution is fixed at 24 km. Conversely, when the SST resolution is downscaled, the SSH resolution is fixed at 80 km.

## 3.2 Impact of sinking velocities and type of inputs

In Figure 4 we compare the prediction metrics in terms of $BL_{200m}$, $F_{200m}$ and the percentage of valid predictions provided by the 3 CNNS (i) $Unet^w_{5V-4L}$, (ii) $Unet^w_{5V-1L}$, and (iii) $Unet^w_{sst-ssh}$. Additionally, the scores obtained with the standard catchment areas, i.e. $box_{200km}$ and $box_{100km}$, were computed. Overall, the scores improved with larger sinking velocities $w$. This is probably because particles with high $w$ are less sensitive to subsurface dynamics and are likely to be much closer to the sediment trap location, making it easier to predict the location. Conversely, with a lower sinking velocity, the particle path is typically more complex, with a longer transit resulting in a catchment area that is typically further from the sediment trap location and spread over a larger area, as shown by Wang et al. (2022a).

A comparison of $Unet^w_{sst-ssh}$ predictions with traditional 100-200 km area baselines (Figure 4) reveals a clear added value of the neural network scheme. The $box_{200km/100km}$ gives on average between 1% and 20% of valid predictions, with the percentage of predicted surface particles averaging about 20%. In contrast, the $Unet^w_{sst-ssh}$ outperforms this score with a percentage of valid predictions ranging from 50% ($w = 80$ m.d$^{-1}$) to 80% ($w = 300$ m.d$^{-1}$). The average percentage of predicted particles $F_{200m}$ increases to 50% with $Unet^w_{sst-ssh}$ (+30% compared to the boxes).

To gain a deeper understanding of the limitations of the $Unet^w_{sst-ssh}$ score, we have increased the dynamical information in the region provided by the inputs using $Unet^w_{5V-1L}$ and $Unet^w_{5V-4L}$. Unlike $Unet^w_{sst-ssh}$, $Unet^w_{5V-1L}$ include explicit surface velocity and vorticity information at a fine resolution of 8 km. This additional information has led to a ∼5-10% increase in valid predictions. As explained in Figure 3, part of this improvement is due to the finer resolution. Thus, the addition of velocity and vorticity does not seem to significantly improve the score prediction at this resolution. We assume that $Unet^w_{sst-ssh}$ can correctly extract the relevant features of the geostrophic velocities directly from the SSH.

The main limitation of the predictions seems to be the lack of information at deep levels. Indeed, $Unet^w_{5V-4L}$ outperforms all other $Unet$ models with a range of 78% to 99% accuracy in predicting particle path dynamics ($F_{200m}$ = 60-80%), suggesting that deep dynamics are a crucial factor to consider in reconstructing the particle path. In the following section, the role of deep dynamics on particles' pathways is elucidated, showing a direct correlation between the prediction score and the intensity of deep currents.

## 3.3 Impact of deep dynamics

The aim of this investigation is to examine the role of the deep dynamics on particle pathways and their potential impact on the $D^w_{simu}$ predictions. Based on the model average kinetic energy profile ($KE = \frac{1}{2}(u^2 + v^2)$) in the region (Figure B1), it seems that the dynamics below 1000 m could be considered negligible compared to the mesopelagic zone (z < 1000 m). Despite the low intensity of the deep currents, the particle pathways are still significantly influenced by deep structures such as deep jets or mesoscale eddies, which can originate from the surface or at depth through local bathymetric interactions (Smilenova et al., 2020). The deep currents induced by the continental shelf clearly affect the movement of the particles as soon as they are released, as shown in Figure 5(a-d) by the PDF of particles (w=100 m.d$^{-1}$) when they reach the mesopelagic layer (1000 m depth). In this example, the particles are already ∼ 100 km away from the source before entering the area driven by surface

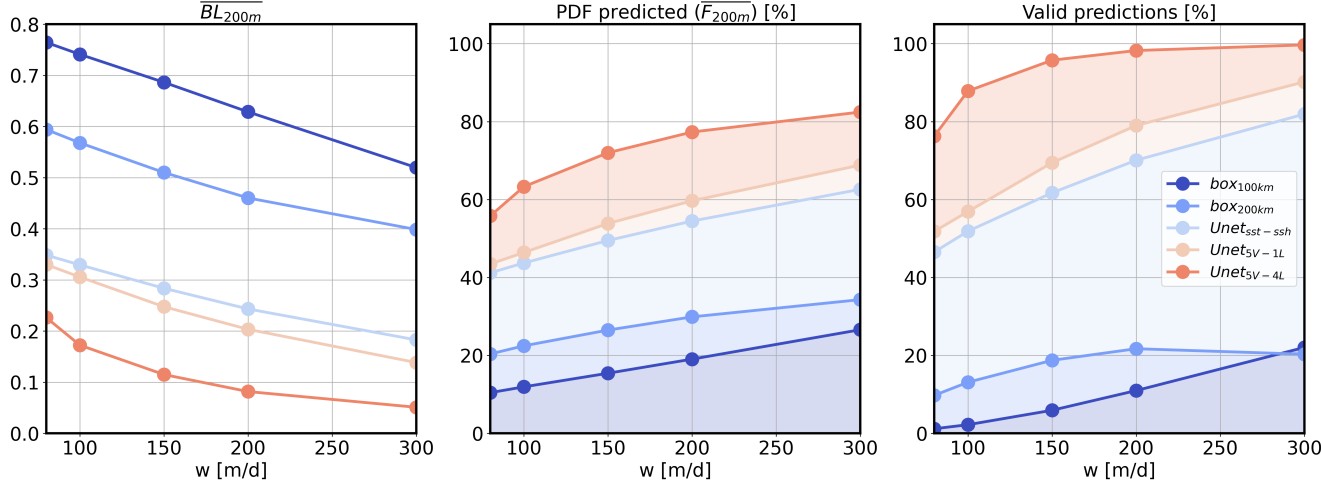

**Figure 4.** Evaluation of 3 types of $Unet$ depending on the sinking speed w. $Unet_{sst-ssh}$ is in light blue, $Unet_{1L-5var}$ in beige and $Unet_{4L-5var}$ in red. Additionally, the scores of $box_{100m}$ (dark blue) and $box_{200m}$ (blue) have been computed. We evaluate the score using (a) $BL_{200m}$, (b) $F_{200m}$ and (c) the % of valid predictions.

conditions. Moreover, based on a comparison between the two KE maps (b) and (d), the local currents around the PAP-SO station (see inside the black box) in the upper layer (0-1000 m) are typically not well correlated with the dynamics at depth (1000 m-3000 m). This leads to an incorrect prediction area (red contours vs black contours). Although, some surface eddies can have very deep coherence. If they are close to the PAP-SO station, they can lead to a coherent connection as they ensure a better correlation between surface and deep dynamics (Figure 5(f,h)). They also tend to trap the particles together. These effects seem to reinforce the predictive power, as exemplified in Figure 5(e).

To corroborate these observations, we analyse the link between the score of the $Unet_{sst-ssh}^{100}$ model and (i) the shape of the "true" particles particle catchment area (i.e., the catchment area from Lagrangian experiments) when reaching the base of the mesopelagic zone (z = 1000 m) (Figure 6a), and (ii) the local deep dynamics (KE and $\zeta$ below 1000 m) (Figure 6b).

For (i) we computed the averaged bin statistics of the prediction score $BL_{200m}$ conditioned on the mass centre and the entropy of the true PDFs at 1000 m. The mass centre is defined as the average distance of the particles from the ST location. The entropy, defined as $-\Sigma p_i log(p_i)$, where $p_i$ is the PDF value at point i, describes the spread of the PDF. A high entropy is associated with a large particle spread over the domain (Picard et al., 2024). This demonstrates that the final prediction score $BL_{200m}$ is directly related to the PDF state at deep depth. It can be observed that valid scores ($BL_{200m} < 0.3$) are associated with a low value of mass centre and high value of entropy. This suggests that particles whose centre of mass remains close to the sediment trap location, even when dispersed over a large area at depth, are competently predicted. Conversely, the particles significantly affected by the deep currents reaching the mesopelagic zone too far away from the PAP-SO ST (mass centre > 75-100 km) are unlikely to be competently predicted.

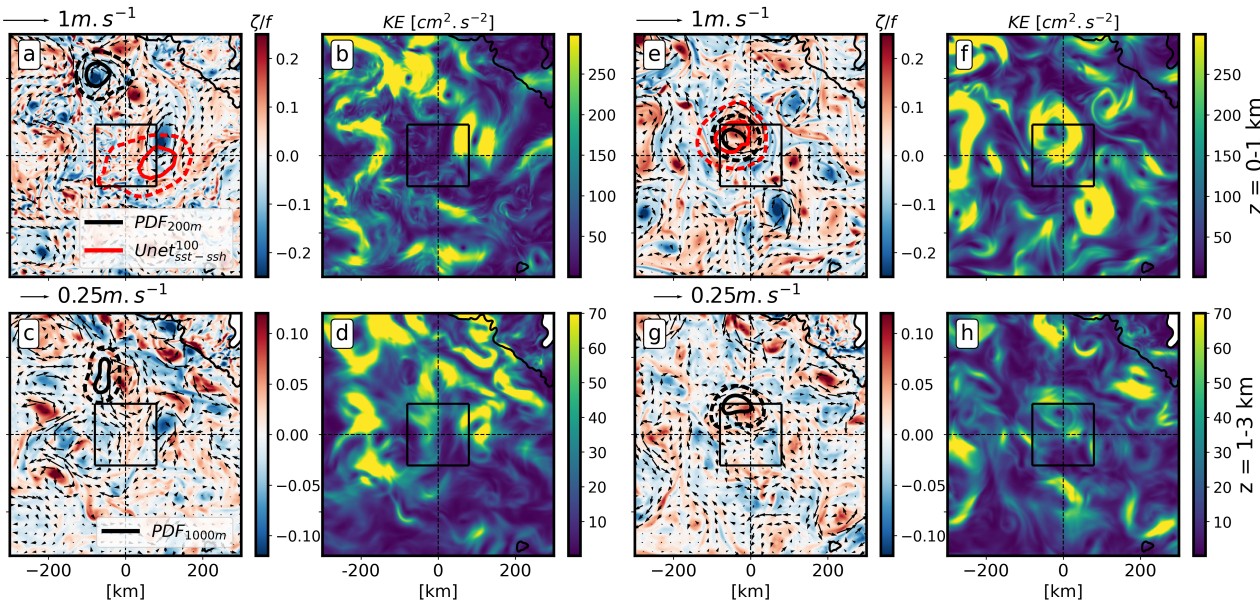

**Figure 5.** Example of predictions from the numerical simulation between 29 Feb and 30 Mar 2004 (a-d) and 6 Sep and 6 Oct 2004 (e-h). (a,b,e,f) Relative vorticity ($\zeta/f$), currents and kinetic energy ($KE$) vertically averaged between 1000 m and 3000 m and temporally averaged during the particles crossing. The "true" particle catchment area at 1000 m is indicated by the black contours, which contain 25% and 75% of the PDF, respectively. (c,d,g,h) Relative vorticity, currents and kinetic energy vertically averaged between 0 m and 1000 m and temporally averaged during the particles crossing. The "true" particle catchment area at 200 m is shown by the black contours and the associated $Unet^{100}_{sst-ssh}$ prediction is shown by the red contours.

For (ii), the averaged bin statistics of the prediction score $BL_{200m}$ were conditioned with the local KE and relative vorticity averaged vertically between 1000 m and 3000 m, horizontally in an 80 km box (black box in Figure 5), and temporally during the crossing of the particle in the layer. A clear indication of a favourable prediction score ($BL_{200m} < 0.3$) can be observed when either the horizontal velocity is weak (i.e. low KE) or the absolute value of the vorticity is high (i.e. presence of a mesoscale eddy). These results corroborate the finding that the deep currents are the primary driver of the final prediction score.

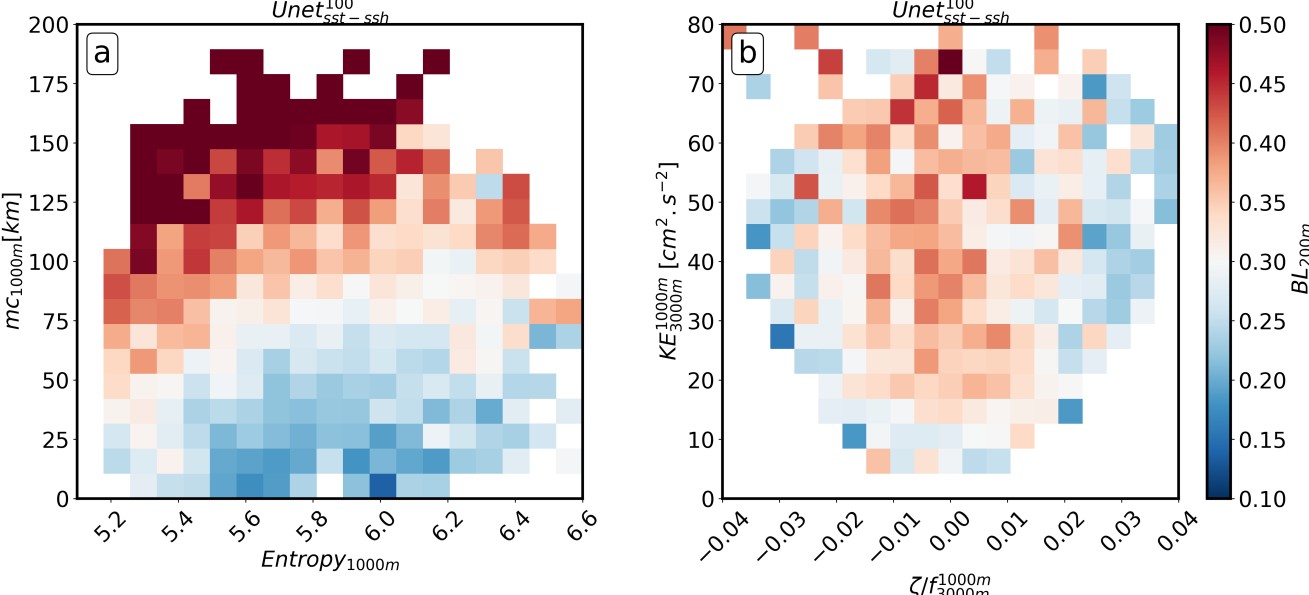

**Figure 6.** Averaged bin statistics of the $Unet_{sst-ssh}^{100}$ prediction score $BL_{200m}$ in the (a) Particle $PDF_{1000m}$ mass centre and entropy (b) kinetic energy (KE) and relative vorticity ($\zeta/f$) averaged between 1000 m and 3000 m and in an 80 km box centred on the sediment trap location and temporally during the particles crossing.

## 4 Connection between surface and deep fluxes observations at the PAP-SO station

This section presents the application of $Unet_{sst-ssh}$ with real satellite-derived observations around the PAP-SO station and examines whether the predicted catchment areas improve the correlation between deep sediment trap fluxes and the surface chlorophyll-a concentration.

### 4.1 Predictions with satellite data

We focus on a 20-year period from 01/01/2000 to 01/06/2019. The data used in this study were obtained from the Global Ocean Gridded L4 Sea Surface Heights And Derived Variables Reprocessed from Copernicus Climate Service, with a resolution of $0.25° \times 0.25°$ (https://doi.org/10.48670/moi-00148), and the Global Ocean OSTIA Sea Surface Temperature and Sea Ice Reprocessed with a resolution of $0.05° \times 0.05°$ (https://doi.org/10.48670/moi-00168). For SST and SSH, we sampled a daily dataset once every 10 days and interpolated the maps over the original CROCO grid in an 800-km box centred at PAP-SO station using a bicubic interpolation method. To ensure coherence between the satellite dataset and the simulation dataset used to train $Unet_{sst-ssh}^{w}$, we compared the SSH and SST distributions between the two datasets and no significant differences were observed (Figure A1). The satellite-derived SSH and SST datasets are used as inputs to generate predictions with $Unet_{sst-ssh}^{w}$. Over the 20-year period (2000-2019), a total of 815 predictions were generated for each sinking velocity $w$, with one PDF

prediction generated every 10 days. We denote the resulting dataset of predicted PDFs as $D_{sat}^w$. To ensure that the predictions produced with satellites are consistent with the predictions observed with the simulation data in Section 3, we compare the respective shape characteristics in $D_{sat}^w$ and $D_{simu}^w$ (mass centre and entropy distribution, Figure A2). No significant differences were found, providing further confidence in the prediction made with real satellite-derived data.

Figure 7 shows examples of catchment area predictions from $D_{sat}^{100}$ between June and October 2016. The PDFs are associated with the corresponding chlorophyll-a images as background and the geostrophic sea surface velocities (averaged over the period). The surface chlorophyll-a images are derived from Global Ocean Colour Plankton and Reflectances MY L3 daily observations at 4 km resolution (https://doi.org/10.48670/moi-00282). The PDFs from $D_{sat}^w$ are associated with a date representing the mean time of particles arrival at the surface. The images illustrate a discernible coherent time continuity between
the $D_{sat}^{100}$ catchment area locations. The $D_{sat}^{100}$ PDFs are often outside the $box_{200m}$ and show narrower locations that can change rapidly, usually in less than a month.

## 4.2   Particle flux data at the PAP-SO station

All particle flux data used in this study are from PAP-SO STs (Lampitt and Pebody, 2023) deployed between 3000 m and 3200 m, which is approximately 1800 m above the seabed (see Lampitt et al. (2010) for a detailed methodology). The collection
period varies between 7 and 42 days, depending on the time of the year and expected fluxes. Fluxes are integrated over the collection period and expressed in $mg\,m^{-2}d^{-1}$. They are further separated into different variables: Dry weight, Particulate Organic Carbon (POC), and Particulate Inorganic Carbon (PIC). Dry weight is the dry mass of the material collected in the sediment trap, POC is the organic carbon retained on a 0.7 um GF/F filter after acidification and PIC content was calculated as the difference between total carbon and POC content (Lampitt et al., 2023).
Figure 8 shows the 20-year time series of fluxes measured at PAP-SO ST, with chlorophyll-a concentration time-averaged for each 10-day period and spatially over a 200 km box centred at PAP. Irrespective of the flux type, a clear signature is observed during the spring bloom for almost every year. This is characterised by a peak in chlorophyll-a concentration, which is followed later by a peak in deep ocean carbon fluxes. The time lag between the chlorophyll-a and the carbon fluxes depends mainly on the time it takes for particles to travel from the euphotic layer to the ST (Stange et al., 2017). Due to the large range of sinking
velocities, $w$, this time lag, $\delta_t$, can vary significantly from days to months.

## 4.3   Assessing connections between the surface ocean properties and deep ocean particle fluxes

To assess the link between the surface ocean and ST carbon fluxes, we apply a methodology similar to that introduced by Frigstad et al. (2015). This strategy is based on the cross-correlation (CC) between the particle fluxes measured at PAP-SO and the surface Net Primary Production (NPP) averaged over the catchment area. The CC score obtained with the predicted
catchment area $D_{sat}^w$ can be compared with the CC references ($box_{100km}$ and $box_{200km}$) to confirm, or not, an improved relationship between sea surface tracers and deep measurements. In this study, we have chosen to use chlorophyll-a concentration derived from ocean color images instead of NPP to work directly with satellite-derived observations due to the large variability in derived NPP products which depends on the choice of the algorithm used (Saba et al., 2011).

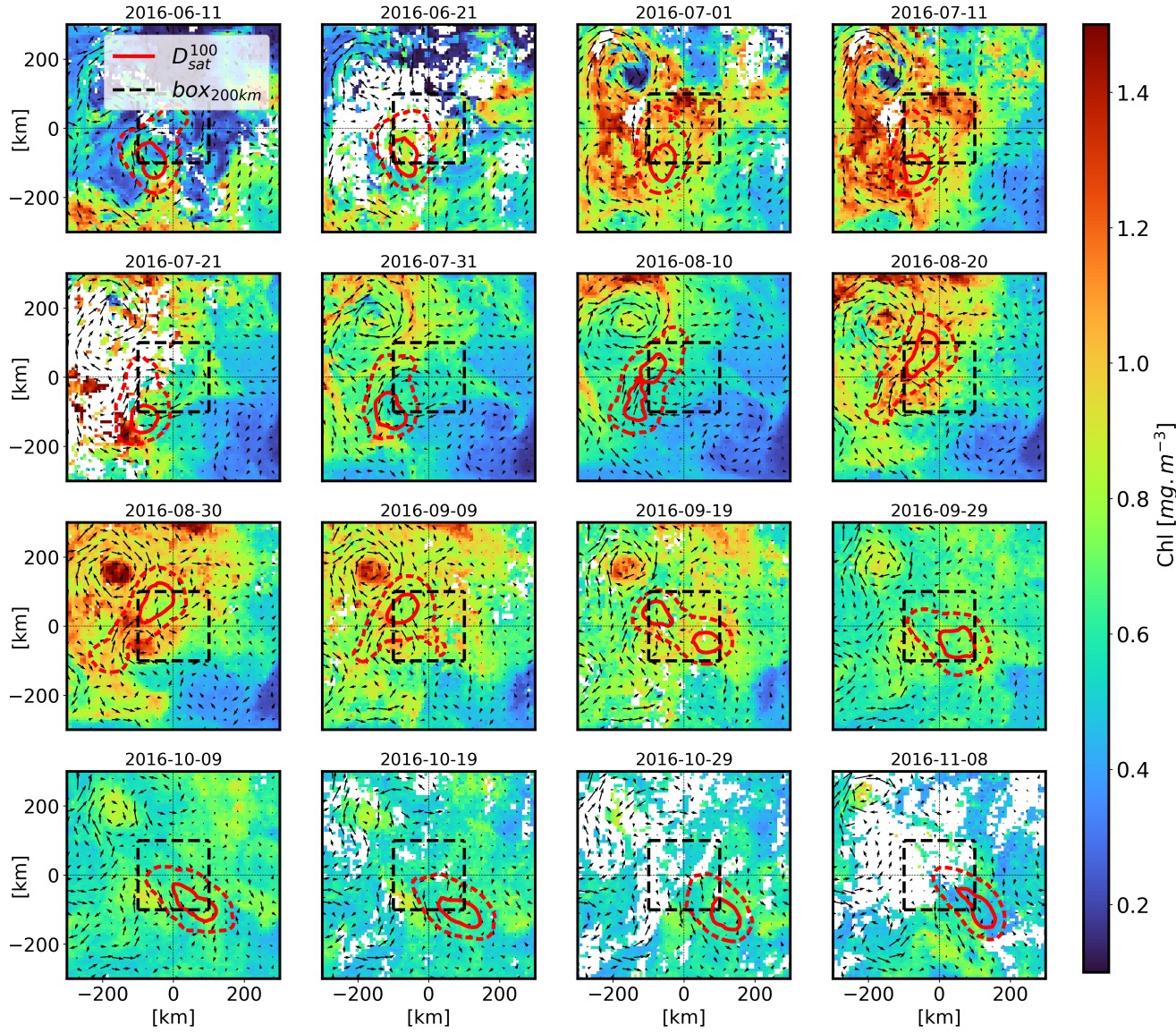

**Figure 7.** Visual results of the $D_{sat}^w$ predictions (predictions with $Unet_{sst-ssh}^{100}$ on real satellite data) represented by the red contours (25% and 75% of the PDF). The plots show the evolution of the PDF from June to October 2016. The 200 km box is shown in black dashed lines. The corresponding chlorophyll-a images from Atlantic Ocean Colour Global Ocean Colour Plankton and Reflectances MY L3 daily observations at 4 km resolution (OCEANCOLOUR GLO BGC L3 MY 009 107) (averaged over a 10-day window). The black arrows represent the geostrophic current from Global Ocean Gridded L4 Sea Surface Heights And Derived Variables Reprocessed Copernicus Climate Service. White areas in the chlorophyll-a data are due to cloud cover.

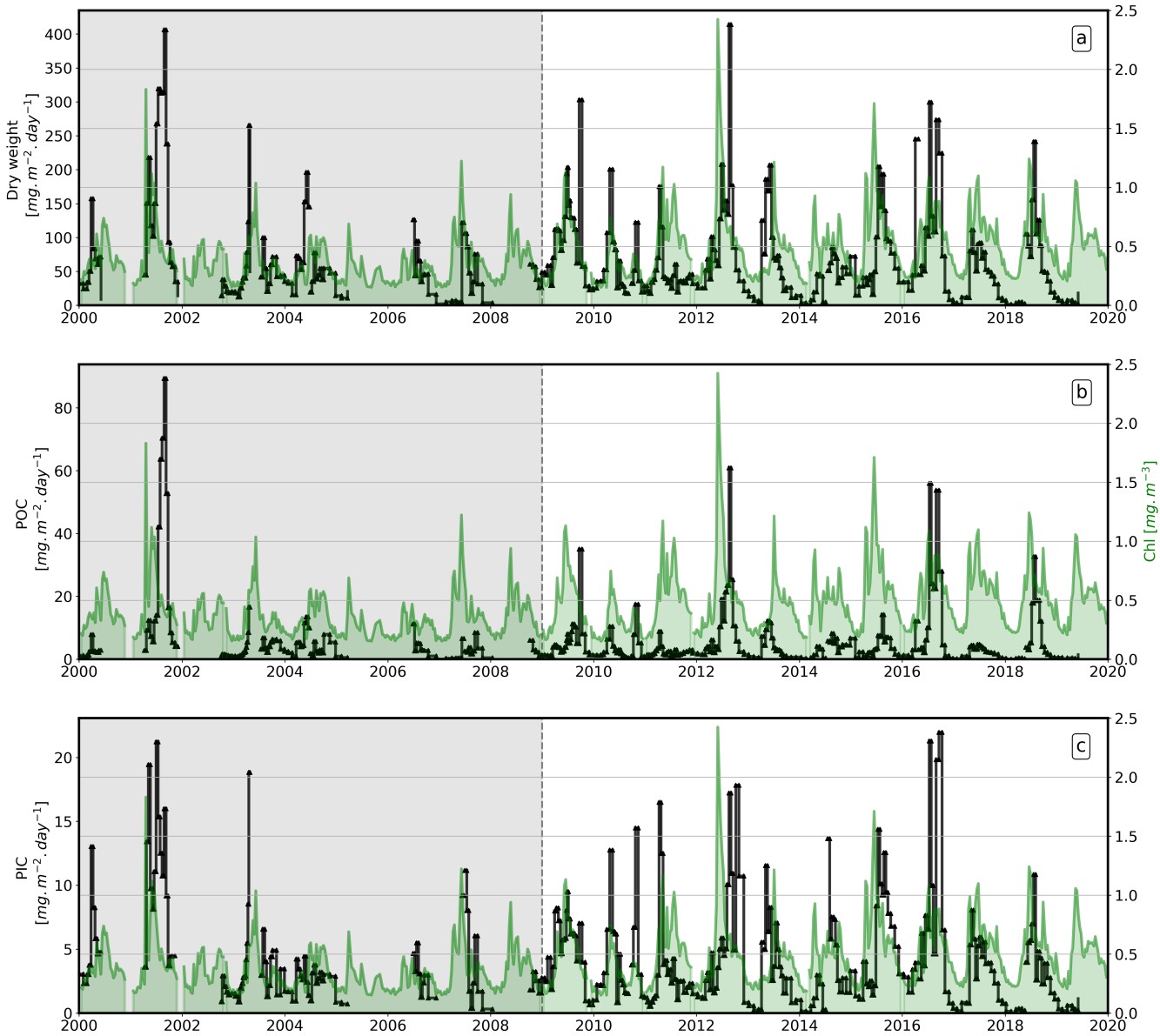

**Figure 8.** Time-series of carbon fluxes (dry weight, Particulate Organic Carbon (POC), Particulate Inorganic Carbon (PIC)) measured at the PAP-SO 3000 m sediment trap between 2000 and 2019 included. The green background is the chlorophyll-a from Atlantic Ocean Colour Global Ocean Colour Plankton and Reflectances MY L3 daily observations (OCEANCOLOUR GLO BGC L3 MY 009 107). The time series are temporally averaged over a 10 days period and spatially in a 200 km box around PAP. The white area represents the data used for the cross-correlation calculation.

The detailed methodology used to compute the CC is described in Appendix B. In summary, the CC is calculated by
determining the correlation coefficient between PAP-SO fluxes (dry weight, POC, and PIC) and the surface chlorophyll-a

concentration within the catchment areas. We associate each particle flux measurement of PAP-SO ST taken at a given time $t$ with averaged chlorophyll-a concentration in the catchment area depending on the time lag $\delta_t$. We compute the CC with 3 types of catchment area which are $box_{200km}$ and $box_{100km}$ (baseline reference) and predictions from $D_{sat}^w$. The sinking velocity considered for the prediction $D_{sat}^w$ depends on $\delta_t$ as defined in Table 1 to account for the variability of the duration of the particle pathways with respect to the sinking velocity. For example, for a time lag of less than 12 days, i.e. $\delta_t < 12$ days , we consider the predictions with the largest sinking velocity w= 300 m.d$^{-1}$ and use the catchment area predictions provided by the $D_{sat}^{300}$ dataset.

To test the robustness of the results, the CC was also computed using random catchment area predictions from the $D_{sat}^w$ dataset. This random process was repeated 100 times to computethe 10th and 90th score percentile for each $\delta_t$, representing the range of uncertainty.

| time lag [days] | $\delta_t > 34$ | $25 \geq \delta_t > 34$ | $18 \geq \delta_t > 25$ | $12 \geq \delta_t < 18$ | $\delta_t < 12$ |
|---|---|---|---|---|---|
| $w(\delta_t)$ | w = 80 m.d$^{-1}$ | w = 100 m.d$^{-1}$ | w = 150 m.d$^{-1}$ | w = 200 m.d$^{-1}$ | w = 300 m.d$^{-1}$ |

**Table 1.** $w(\delta_t)$ predictions as a function of time lag in days. The time lag represent the time for a particle at $w$ velocity to travel from the euphotic layer to the ST.

As a considerable number of ST data points were missing prior to 2009 (Figure 8) and so we only compute the CC between 2009 and 2019 (white area in Figure 8), which is the period where the time series is continuous and considered valid by (Lampitt et al., 2023). A second period between 2009 and 2019, but excluding the years 2011 and 2013, is also examined. The years 2011 and 2013 are associated with the deep fluxes that occur before the chlorophyll-a bloom. This pronounced anomaly has been observed before, and a possible explanation is that rapid re-stratification and/or intense events such as storms isolate pre-bloom particles at depth, leading to an intense carbon export that is not associated with surface data (Giering et al., 2016). Therefore, these years should be filtered out as they are not consistent with the hypothesis of biological processes at the sea surface as the main drivers of deep ocean particle fluxes.

## 4.4 Results

The cross-correlation (CC) was computed at 3-day intervals between $\delta_t = 0$ days and $\delta_t = 110$ days, which is the range in which a non-zero correlation signal can be observed (Figure 9). For both periods, the signal generally peaks at $\delta_t \sim 30 - 20 days$ (w = 100-150 m.d$^{-1}$) for dry weight and PIC. Whereas, POC shows a maximum at $\delta_t \sim 70$ days (w = $45 m.d^{-1}$). The correlations are generally weak for the three particle flux variables and the reasons for this are discussed in the next section. Overall, the CC with the three catchment areas considered has a higher score than the random catchment area zone (blue area), confirming its relevance. However, the score appears to be generally higher for $Unet_{sst-ssh}^w$ predictions, particularly for time lag values of $15 < \delta_t < 50$, which is associated with the particle velocities considered in this study. Note that for $\delta_t > 50$, we choose to continue using $Unet_{sat}^{80}$. However, the associated particle sinking velocity should be slower than the values considered here (w $\leq$ 80 m.d$^{-1}$). Consequently, the $Unet_{sat}^{80}$ may not be optimally suited to this context, which may partly explain why the

correlation improvement is less pronounced here compared to $box_{100km}$ and $box_{200km}$. The period without 2011 and 2013 leads

to a higher global correlation score for all variables. Interestingly, a significantly higher score with $Unet^w_{sst-ssh}$ is observed for PIC at about $\delta_t = 30$ days (w = 100 m.d$^{-1}$), corresponding to the average particle sinking velocity observed in the region. This seems to confirm that the model improves our ability to link surface data with deep carbon fluxes, especially for PIC fluxes.

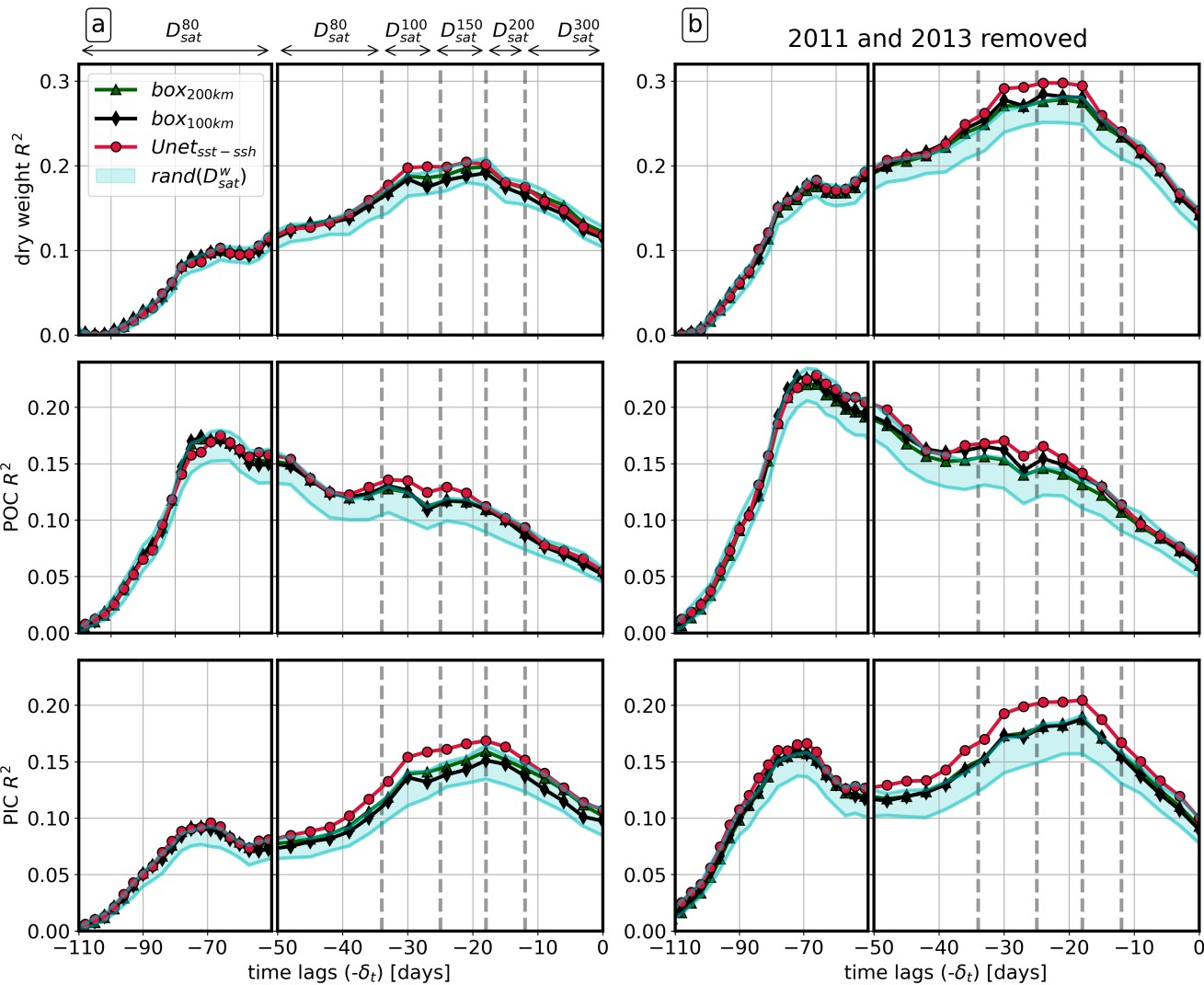

**Figure 9.** Cross-correlations (CC) computed with dry weight, Particulate Organic Carbon (POC), Particulate Inorganic Carbon (PIC) considering the period 2009-2019 (a) and the period 2009-2019 excluding the years 2011 and 2013 (b). CC is computed considering catchment areas from $box_{200km}$ (green), $box_{100km}$ (black) and $D^w_{sat}$ (red). The $D^w_{sat}$ catchment areas depend on the time lag $\delta_t$ as delimited by the dashed lines. The blue area represents the zone between the 90-10th percentile CC score computed with 100 random catchment areas from $D^w_{sat}$.

## 5  Discussion

### 5.1  Comparison with previous studies

There are still significant gaps in the understanding of the PAP-SO ST fluxes, and no discernible link with the surface ocean data (i.e. NPP) has been established (Lampitt et al., 2023). A crucial missing piece of information that may limit such a link is the source location of the particles, which can vary rapidly and be located hundreds of kilometres from the trap, depending on the local surface mesoscale dynamics (Wang et al., 2022a). Conventionally, this dynamic effect has been addressed by considering a fixed zone of influence, typically represented by a 100 or 200 km box surrounding the sediment trap (Lampitt et al., 2023).

However, this approach is limited in its ability to handle the spatial and temporal variability of the source area, which can vary on a weekly basis due to local mesoscale dynamics. The objective of this study is to examine the potential of using machine-learning and surface ocean mesoscale dynamics data to establish a more robust relationship between the deep ocean carbon fluxes from the PAP-SO ST and surface ocean dynamics. This approach has the capacity to make effective predictions, thereby improving the source area location compared to a simple box. The strategy is based on a machine learning framework

described in Picard et al. (2024), where convolutional neural networks were trained with a series of Lagrangian experiments in a numerical simulation to predict the catchment area of a PAP-SO ST. This approach suggested effective predictions using only surface data. Consequently, we developed an extended version of the network, called $Unet^w_{sst-ssh}$, to identify catchment areas at PAP-SO with remote sensing observations. The cross-correlation methodology, based on Frigstad et al. (2015), was used to determine the relationship between surface and deep fluxes. Despite notable differences in methodology compared to

the Frigstad et al. (2015) study, which obtained catchment areas by using particle backtracking in a reanalysis model with a constant sinking rate of $w = 100$ m.d$^{-1}$ and computed the cross-correlation score with the NPP during the period 2006-2016, we found some coherence with our results. Indeed, the observation of a correlation peak for dry weight and PIC at $\delta_t \sim -20/30$ days is in line with Frigstad et al. (2015), who also identified a maximum correlation at PAP-SO for dry weight at $\delta_t = 1$ month. Furthermore, the POC correlation peak occurred at a greater time lag ($\delta_t = 70$ days), which is also supported by Frigstad

et al. (2015), who observed a $\delta_t$ = 2-3 months. Nevertheless, while the results of this study demonstrate the advantages of $Unet^w_{sst-ssh}$, the correlation signal remains weak ($R^2 < 0.3$). This can be partly explained by the fact that not all biological surface drivers have been fully captured by the methodology employed. Additionally, the $Unet^w_{sst-ssh}$ prediction presents limitations due to the absence of information at depth.

### 5.2  Other biological surface drivers of deep ocean particle fluxes

A notable constraint of the study lies in the simplified representation of the organic particles within the numerical simulation. As mentioned in Picard et al. (2024), the size of the particles, and their sinking rate vary during their descent through the water column, due to aggregation/ disaggregation, grazing, remineralization by bacterial activity (Alldredge and Gotschalk, 1988; Berelson, 2001; Fischer and Karakaş, 2009; Villa-Alfageme et al., 2016). These processes have not been taken into account in the presented Lagrangian experiments, and it is clear that they must be considered in future experiments. One potential approach

to achieve this would be to use a Lagrangian framework that incorporates the parameters of particle biological interactions, as proposed by (Jokulsdottir and Archer, 2016).

An other major limitation of this study, particularly with respect to the cross-correlation method, is the simplified assumption that the sinking particles captured by the STs are directly derived from the chlorophyll-a observed at the surface. First, sinking particles do not systematically originate from chlorophyll-a concentration footprints, a proxy for phytoplankton biomass.

According to Nowicki (2022) and Siegel et al. (2023), zooplankton contributes a significant fraction of the sinking export in the region (>50%). In particular Lampitt et al. (2009, 2023) also propose that deep carbon sequestration at the PAP-SO site could be controlled by Rhizaria, which includes two main classes, Radiolaria (mixotrophs) and Foraminifera (heterotrophs). Following the occurrence of phytoplankton blooms, zooplankton converts phytoplankton biomass and detritus into faecal pellets, which facilitate the export and rapid sinking of POC (Steinberg and Landry, 2017). However, zooplankton dynamics were

not explicitly addressed in this study. The CC methodology focuses on the linkage between surface chlorophyll-a concentration and deep ocean particle fluxes and does not account for the contribution of the zooplankton-mediated particle transformation of deep ocean particle flux (Briggs et al., 2020). Zooplankton dynamics likely introduce an additional time lag in carbon export, which may account for the delayed correlation peak observed for POC. Conversely, PIC may be more directly driven by the sinking of phytoplankton-derived calcite incorporated into aggregates or zooplankton-derived calcium carbonate shells which

can sink rapidly (up to 700 m.d$^{-1}$) from the surface, and may explain why the $D_{sat}^{w}$ CC score is the most optimal with this flux (Schmidt et al., 2014). To more accurately explain the drivers of POC pulses to the deep ocean, it would be beneficial in the future to have a more comprehensive representation of zooplankton dynamics in the upper ocean. While being challenging, this topic has been addressed by recent studies in the California Current System, where the zooplankton growth evolution as well as their surface 2D advection have been accurately depicted (Messié and Chavez, 2017; Messié et al., 2022).

A further limitation of focusing only on chlorophyll-a concentration is that it represents the production of phytoplankton organic matter without any species information. However, numerous studies have indicated that the carbon export efficiency is linked to particle characteristics such as the size, density, and sinking velocity which are primarily determined by phytoplankton communities (Henson et al., 2012, 2015). In the future, it would be necessary to refine our analysis by taking into account the local plankton communities observed in the catchment areas to include further information such as the sinking

rate and the export ratio. For instance, the use of OC-CCI micro/nano/pico phytoplankton data (Copernicus Marine Service, daily, 4 km resolution) could facilitate a more comprehensive assessment of the impact of community composition on fluxes, thereby improving our interpretation of the ST data. Moreover, pigment signatures (anomalies in the sea colour signal) are now beginning to be used to map the distributions of dominant phytoplankton groups (Alvain et al., 2005, 2006; Cetinić et al., 2024). More recently, machine learning products used a data-driven approach to extrapolate surface plankton communities (El

Hourany et al., 2019) and biological properties (Sauzède et al., 2017) in the water column from surface conditions and in situ profiles (BioGeoChemical-Argo; Claustre et al. (2020)). Products now available globally and at high resolution (e.g. 8-day product and 4 km for El Hourany et al. (2019) products) may support expanding the variables considered in future work.

A final limitation lies in the limits of the satellite product itself, which only provides an estimate of phytoplankton chlorophyll-a concentration to a maximum of 10 m deep (Wang et al., 2022b). However, the deep chlorophyll-a maximum (DCM) can differ

significantly from the surface state, particularly in oligotrophic conditions with a shallow mixed layer, where the DCM is typically observed down to a maximum depth of 200 m (Mignot et al., 2014). However, this can also be a challenge during the pre-bloom phase. The rapid deepening of the mixed layer depth, followed by a rapid re-stratification (typically during a storm event), can result in the isolation of a significant amount of carbon from the surface (Giering et al., 2016), a phenomenon known as the mixed layer pump (Dall'Olmo et al., 2016). This phenomenon may explain the anomalies observed in 2011 and 2013, where the peak of ST fluxes occurred before the onset of the chlorophyll-a bloom, resulting in a disruption of the global CC score. Further research is therefore required to gain a deeper understanding of the impact of these mechanisms. Emerging technologies, especially BGC-Argo in-situ observations and machine-learning-based products that can be used to estimate the carbon vertical distribution of organic carbon from satellites (Sauzède et al., 2016), are likely to be of key interest. Some of these products (i.e, 3D fields of Particulate Organic Carbon, Particulate Backscattering coefficient and Chlorophyll-a concentration) are already available (https://doi.org/10.48670/moi-00046). They could also provide a more comprehensive assessment of the missing NPP obtained from a surface-only perspective.

In the future, it would be beneficial to extend the catchment area reconstruction to other long-term ST observation sites which cover different regions and systems in the global ocean e.g. BATS (Bates and Johnson, 2023), Station M (Smith et al., 2018), DYFAMED (Miquel et al., 2011), and ALOHA (Howe et al., 2011). Hence, the integration of the aforementioned processes in the proposed machine-learning methodology seems to be a relevant research avenue to generalize beyond station-specific characteristics and to provide a more comprehensive record of deep ocean carbon fluxes.

## 5.3   The importance of representing deep ocean dynamics

It is important to consider this analysis in the context of the previous study by Picard et al. (2024), which considered PAP-SO ST at a depth of 1000 m with a particle sinking velocity of w=50 m.d$^{-1}$. Despite the relatively low particle velocities, the scores of $Unet_{5V-1L}$ obtained in the aforementioned study were considerably higher (85% of valid predictions, i.e. $BL_{200m} > 0.3$) than those observed in the present study. It was originally hypothesised that the weak deep ocean dynamics at PAP would result in the particle sinking velocity being the primary factor influencing the prediction score. However, we can hypothesise that the use of a comparable sinking speed of 50 m.d$^{-1}$ in this study would result in less than 50% of valid prediction (considering that the score decreases with lower w and that at the slowest sinking rate of 80 m.d$^{-1}$ we only reached $\sim$ 50% of valid predictions with $Unet_{5V-1L}$). Hence, this study seems to indicate that the vertical distance from the upper ocean and the resulting influence of local deep dynamics may be more important than initially hypothesised. Indeed, our results show that the prediction score is significantly driven by the local deep dynamics below 1000 m (Figure 6). As noted by Bolton and Zanna (2019), machine learning faces challenges in reconstructing currents below a certain depth, even in the absence of topography, largely due to the influence of bottom drag. These difficulties are exacerbated when topography is present, as geostrophic currents interacting with the seafloor generate strong bottom-intensified currents that can extend thousands of metres into the water column without leaving a detectable surface signature (e.g. Carli et al., 2024). In addition, submesoscale coherent vortices generated on nearby seamounts, ridges, and continental slopes (Smilenova et al., 2020) can generate anomalous midwater column currents, contributing to the complexity of current structures that cannot be captured without local measurements.

As a result, the primary limitations of $Unet^w_{sst-ssh}$ can be attributed to the lack of comprehensive data on deep currents: the comparison between $Unet_{5V-1L}$ and $Unet_{5V-4L}$ outlines a potential $F_{200m}$ score increase of $\sim +20\%$ with the addition of information at depth (Figure 4).

Hence, the predictive capabilities of $Unet^w_{sst-ssh}$ could be improved by incorporating in-situ observational data into the inputs. To achieve this, data on deep currents will need to be provided, for example by incorporating data from current meters deployed at the PAP-SO ST mooring. An alternative approach would be to focus on the specific sampling period of the recent PAP observation campaigns, during which in-situ drifting sediment traps were released in the mesopelagic (i.e. during the APERO campaign, where 10 drifters were released between the surface and 1000 m for 5 days and the data presented in Baker et al. (2020). However, the limited spatial resolution of the data in the region may prove insufficient to achieve the desired improvement in prediction score. It would be beneficial to conduct a prior study to evaluate the sensitivity score with deep data to determine if such data could improve $Unet^w_{sst-ssh}$. If this approach proves ineffective, an alternative idea would be to consider deploying sediment traps at shallower water depths, but deeper than 1000 m. Previously, STs have been deployed at the PAP-SO site at 1000m but the measurements are more susceptible to under-collection due to hydrodynamic biases associated with conical STs, as highlighted in previous studies (Buesseler et al., 2007; Baker et al., 2020). It is also possible to consider the deployment of sediment traps in a region where the deep dynamics are even weaker and unaffected by the near-topography, which is typically the source of deep eddies and instabilities (Smilenova et al., 2020). A numerical simulation such as the one used here can be used to identify the weakest dynamical regions, where particle pathways below the mesopelagic layer are unlikely to be affected. Nevertheless, 3D numerical simulation remains one of the most effective methods for studying deep ocean dynamics and further efforts are required to validate the accuracy of simulations of deep ocean currents. In addition, given that SSH is the main driver of the network score (see Figure 3), we hypothesise that the network relies predominantly on geostrophic currents to perform its prediction. Consequently, it would be worthwhile to compare the efficiency of the model in regions with varying degrees of geostrophic current dominance.

Finally, questions remain about the uncertainties associated with the Lagrangian method. It is clear that the uncertainty associated with the Lagrangian method has a direct impact on the predictions, since the network is trained directly with the backtracked particles. Although sensitivity tests have been carried out (changing the number of particles and the size of the released patch) to ensure that the particle sources are not affected, some diffusion processes are not represented in the numerical simulation and consequently in the propagation of the particles. To evaluate potential biases, it would be necessary in the future to adopt a stochastic approach (Mínguez et al., 2012), where random noise is introduced into the particle trajectories to account for subgrid scale diffusion processes. These processes have the potential to influence the results of the Lagrangian analysis (see Appendix D for an example). Consequently, the diffusion parametrization should be carefully defined, taking into account local dynamics. This approach would facilitate the establishment of a confidence interval for the source areas.

## 6 Conclusions

This study presents a novel machine learning tool, named $Unet_{sst-ssh}$, which is capable of predicting the catchment area of particles trapped at the PAP-SO station ST moored at 3000 m depth, based solely on remote sensing data, namely SST and SSH. The study considers five sinking velocities, ranging from 80 m d$^{-1}$ to 300 m d$^{-1}$. The results of our method are compared with the direct use of a 100-200 km box around the trap location, representing the conventional approach of catchment area. The results show that the prediction score increases with $w$, and while the 100-200 km boxes predict only 20-30% of the particles catchment area (w=80-300 m.d$^{-1}$), the $Unet_{sst-ssh}$ predictions enhance this score to 40-60%. We applied $Unet_{sst-ssh}$ to real satellite observations at PAP-SO, resulting in the generation of a 20-year catchment area dataset available at a 10-day resolution. The dataset demonstrated a stronger correlation, and therefore connection, between the deep ocean particle fluxes measured at PAP-SO and surface chlorophyll-a concentration compared to the traditional catchment area method. The presence of deep ocean energetic dynamics that are uncorrelated with the surface appears to be the main reason for the invalid predictions. Future improvements to the $Unet_{sst-ssh}$ method would entail a more comprehensive consideration of these deep currents. Ultimately, the improved identification of the surface catchment area of particles collected in deep ocean sediment traps would facilitate the identification of the surface drivers of deep ocean carbon sequestration, thereby improving our understanding of the biological carbon pump.

*Code availability.* The codes used in this study are available online at https://github.com/TheoPcrd/SPARO (last access: October 7, 2024 (Picard, 2024b).)

*Data availability.* The dataset of the predicted catchment area at PAP-SO ($D_{sat}^w$) is available online at https://doi.org/10.17882/102535 (Picard, 2024a)

*Video supplement.* A video abstract is available at https://doi.org/10.5281/zenodo.10261827 (Picard, 2023).

## Appendix A: Statistical comparison between predictions with satellite and simulation database

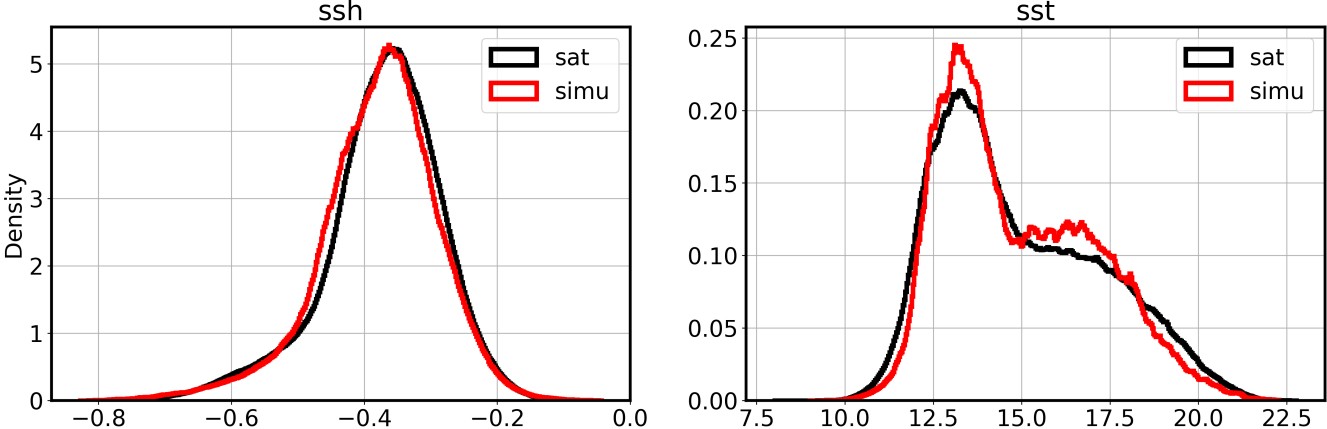

**Figure A1.** Comparison of $Unet_{sst-ssh}$ inputs (SST and SSH distribution) from the satellite data (in black) and from the training dataset from the CROCO numerical simulation (in red).

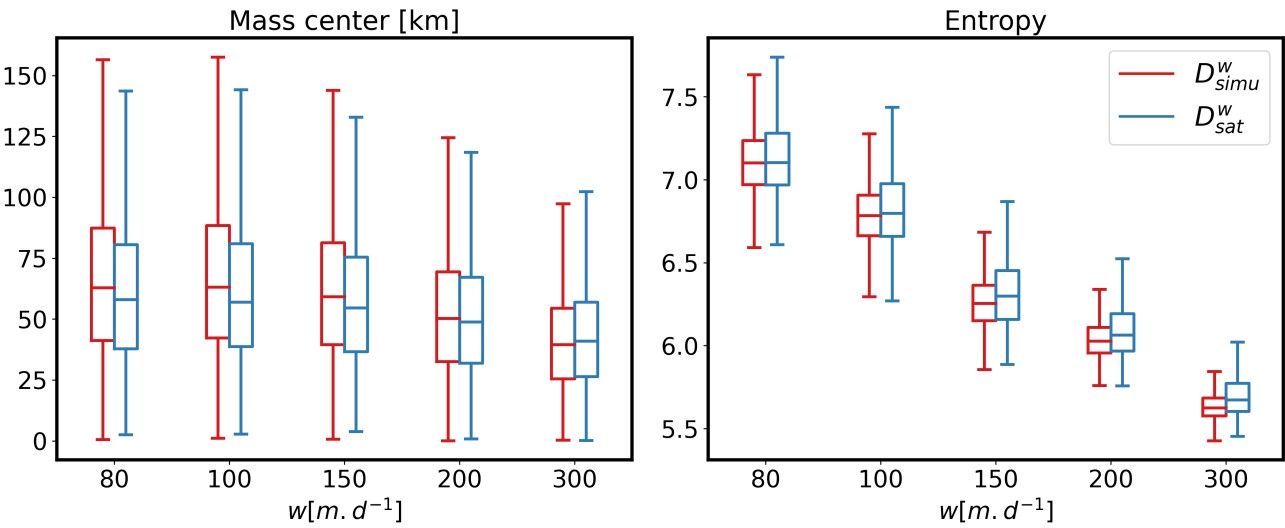

**Figure A2.** Statistical comparison of the catchment area PDF's mass centre and entropy between predictions with numerical simulation inputs $D_{simu}^{w}$ (red) and predictions with satellite inputs $D_{sat}^{w}$ (blue) for different vertical sinking velocities w. The box plot represents the 1st and the 3rd quartiles.

## Appendix B: Dynamics profile at PAP-SO station

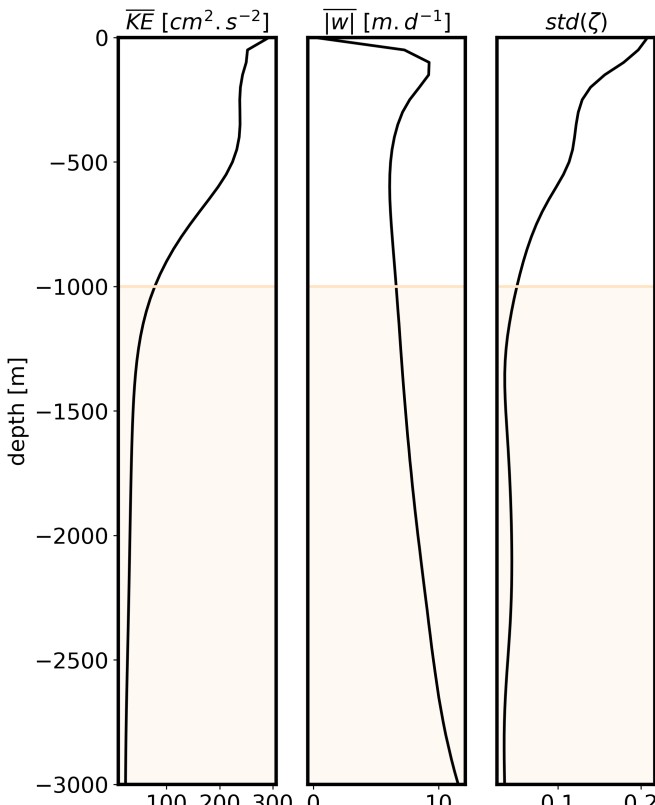

**Figure B1.** Profiles of the average kinetic energy $\overline{KE}$, absolute vertical velocities $\overline{|w|}$ and vorticity standard deviation std($\zeta$) between the surface and 3000 m depth. The profiles are spatially averaged in an 80 km box around PAP-SO station and temporally during 8 years in the POLGYR numerical simulation.

### Appendix C:  Methodology for the cross-correlation calculation

We associate each measurement taken at the PAP-SO ST at a middle time $t$ (in day) and between $t_{start}$ and $t_{end}$ with a corresponding surface chlorophyll-a product depending on (i) the collection period $cp = t_{end} - t_{start}$ and (ii) a time lag $\delta_t$ which represents the time of the particle travelling from the euphotic zone to the ST depth. Since this travelling time depends on particle sinking speed, we decide to assign a time lag range corresponding for each velocities considered in this study (Table 1). As a result, the final PDF prediction area prediction for the time $tl = t - \delta_t$, called $\overline{D_{sat}^{w(tl)}(tl)}$, will also depends on $t_{start}$ and $t_{end}$ such as :

$$\overline{D_{sat}^{w(\delta_t)}(tl)} = < D_{sat}^{w(\delta_t)} >_{tl_{end}}^{tl_{start}} \tag{C1}$$

Where $< . >_{tl_{end}}^{tl_{start}}$ is the average of all the predicted catchment areas between the time $tl_{start} = t_{start} - \delta_t$ and $tl_{end} = t_{end} - \delta_t$.

Similarly, we compute a chlorophyll-a background based on a L3 daily product from Atlantic Ocean Colour Global Ocean Colour Plankton and Reflectances MY L3 daily observations (OCEANCOLOUR GLO BGC L3 MY 009 107). The ocean colour images have been also interpolated over the CROCO grid using linear interpolation. The associated weighted averaged

surface chlorophyll-a background $\overline{Chl(tl)}$ is computed such as :

$$\overline{Chl(tl)} = < Chl >_{tl_{end}+5}^{tl_{start}-5} \tag{C2}$$

To avoid important cloud coverage, particularly during short collection time, we consider 10 additional days during the averaging process (5 days before $tl_{start}$ and 5 days after $tl_{end}$). Finally, we compute the average chlorophyll-a inside the catchment area PDF predicted for time $tl$:

$$\overline{Chl_D(tl)} = \overline{Chl(tl)} \times \overline{D_{sat}^{w(\delta_t)}(tl)} \tag{C3}$$

The comparison is made by computing the average chlorophyll-a inside the reference catchment area boxes, $box_{100km}$ and $box_{200km}$, such as :

$$\overline{Chl_{box200}(tl)} = \overline{Chl(tl)} * box_{200km} \tag{C4}$$

$$\overline{Chl_{box100}(tl)} = \overline{Chl(tl)} * box_{100km} \tag{C5}$$

## Appendix D: The impact of diffusion process on Lagrangian experiments

To illustrate the effects of subgrid scale diffusion processes on particle trajectories, we have implemented a simplified Markov model (Berloff and McWilliams, 2003) of order 0. The computation of the particle trajectory $x_n$ can thus be described as follows:

$$x_{n+1} = x_n + \Delta t \cdot u(x_n, t_n) + \mathcal{R}\sqrt{(2 \cdot K_{diff} \cdot \Delta t)} \tag{D1}$$

Here, the $u$ function is employed to compute the advection of the particles. The final term is related to the stochastic implementation, where $\mathcal{R} = \mathcal{N}(0,1)$ denotes a random number selected according to a normal distribution $K_{diff}$ represents the diffusivity coefficient, and $\Delta t$ is the online step time set to 120 s. The following illustrative examples demonstrate how the

catchment area can be affected by adding a constant diffusivity term. The examples present a period of unfavorable conditions
in winter, characterised by chaotic flows. We focus on the catchment areas PDF observed for two distinct values of $K_{diff} = 0.1$
$\mathrm{m^2s^{-1}}$ (Figure D1) and $K_{diff} = 1$ $\mathrm{m^2s^{-1}}$ (Figure D2), which correspond roughly to horizontal diffusivities associated with
internal waves and submesoscale processes at scales 0.1 to 10 km (Garrett, 1983; Ledwell et al., 1998; Nencioli et al., 2013).
For each value of $K_{diff}$, 10 Lagrangian experiments have been conducted and an averaged PDF of this experiments has been
computed and compared with the Gaussian-filtered PDF that has been used in this study.

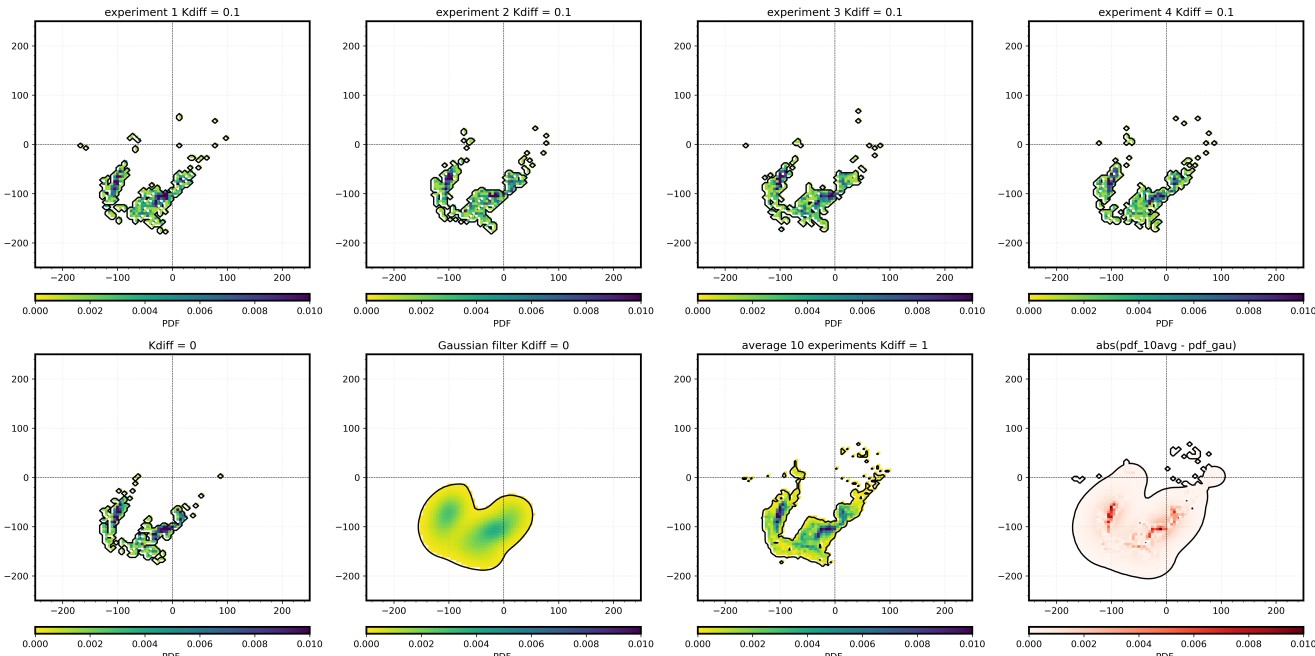

**Figure D1.** First row : 4 examples of PDF with $K_{diff} = 0.1$ $\mathrm{m^2s^{-1}}$. Second row respectively from left to right : PDF for the same period
with $K_{diff} = 0$ $\mathrm{m^2s^{-1}}$, PDF $K_{diff} = 0$ $\mathrm{m^2s^{-1}}$ after gaussian filter, average of 10 PDF with $K_{diff} = 0.1$ $\mathrm{m^2s^{-1}}$, absolute error between the
two previous PDFs.

As demonstrated by the example, when $K_{diff}$ is set to 0.1, the high-density areas of the averaged PDF appears to be included
in the gaussian-filtered PDF (Figure D1). However, when increasing the $K_{diff}$ to 1, the averaged PDF is distributed over a
larger domain and new potential source areas outside the gaussian-filtered PDF can be revealed (see the new particle patch on
the top right (Figure D2).

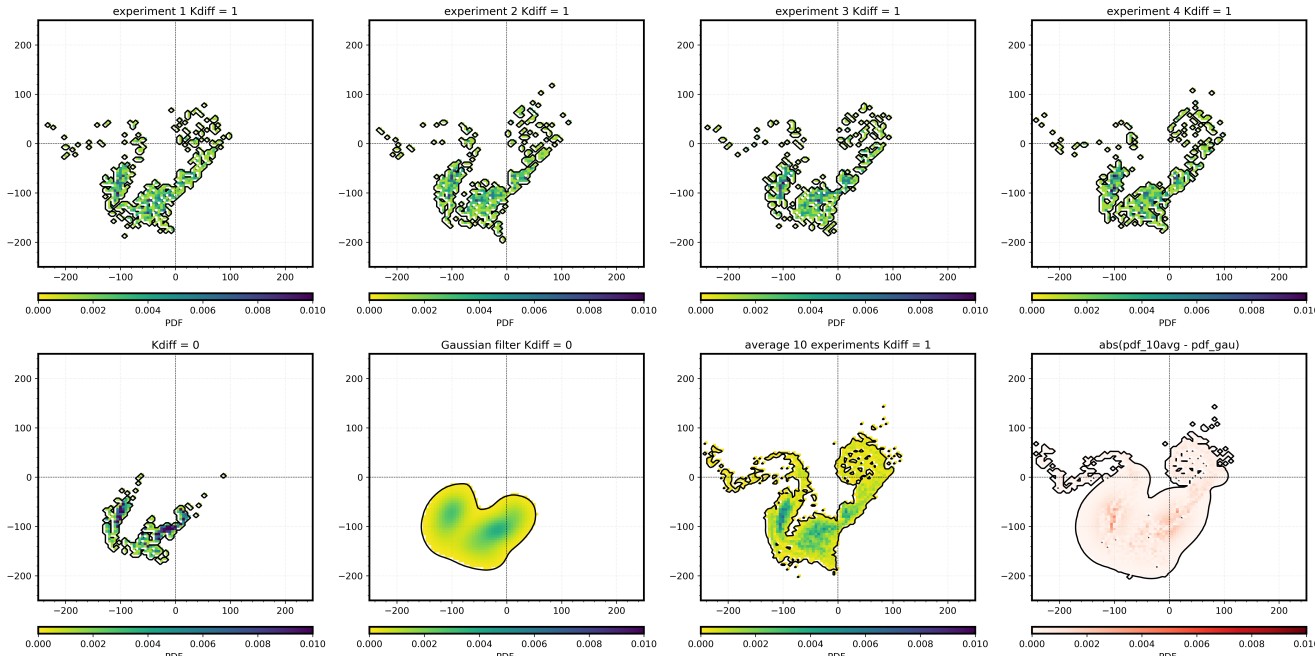

**Figure D2.** First row : 4 examples of PDF with $K_{diff} = 1$ m$^2$s$^{-1}$. Second row respectively from left to right : PDF for the same period with $K_{diff} = 0$ m$^2$s$^{-1}$, PDF $K_{diff} = 0$ m$^2$s$^{-1}$ after gaussian filter, average of 10 PDF with $K_{diff} = 1$ m$^2$s$^{-1}$, absolute error between the two previous PDFs.

*Author contributions.* TP conducted the analysis and prepared the paper with contributions from all co-authors.

*Competing interests.* The authors declare no competing interests.

*Acknowledgements.* We would like to thank Corinne Pebody for her support with access to and understanding the PAP-SO sediment trap data, Mathieu Le Corre for providing CROCO simulation outputs and Monique Messié for valuable discussions. T.P. was supported by a CLASS ECR fellowship and C.A.B and R.L. were funded by the CLASS project (NERC grant NE/R015953/1). The authors would like to acknowledge support from the French National Agency for Research (ANR) through the project DEEPER (ANR-19-CE01-0002-01) and AI 530   chair OceaniX (ANR-19-CHIA-0016). Simulations were performed using HPC resources from GENCI-TGCC (Grants 2022-A0090112051), and from HPC facilities DATARMOR of "Pôle de Calcul Intensif pour la Mer" at Ifremer Brest France.

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
