# Peer review of "Estimating the variability of deep ocean particle flux collected by sediment traps using satellite data and machine learning"

_EGUsphere, 2024_

## Author Comment (AC1)

We thank the reviewers for their positive and constructive comments. We have addressed all comments as detailed below.

**R2**

**Comments on the Manuscript :** Estimating the variability of deep ocean particle flux collected by sediment traps using satellite data and machine learning

**General assessment:** This manuscript investigates the use of a machine learning approach (CNN/Unet) to determine the catchment area of sinking particles measured by deep sediment traps at 3000 m depth at the PAP site, evaluating the influence of particle sinking velocities and using only surface satellite data (SST and SSH). The approach is trained and evaluated using a Lagrangian particle tracking model, and subsequently applied to satellite observations to produce a 20-year dataset. The study builds on a previously published methodology (Picard et al., 2024), where a similar framework was developed for 1000 m traps.

Overall, I find the study well-written and the results convincing. It addresses an important topic in the ocean carbon cycle and particle flux dynamics. The work contributes significantly to improving our understanding of vertical export processes in the ocean, particularly the spatial origin of sinking particles. I support publication after consideration of the following comments.

**Major Comments:**

1. **Training data origin – Model vs. Satellite data:** A key point to clarify is the choice of training the Unet SSH-SST model using model-derived surface data, while the final objective is to apply it to satellite-based observations. Given that the authors themselves acknowledge the limitations in the reconstruction of mesoscale dynamics in numerical models, it would be important to further justify this methodological choice. Why was the Unet model not directly trained using satellite data from the outset, especially since satellite-derived SSH and SST are readily available? Such an approach could potentially reduce the risk of propagating simulation-specific biases into the model predictions.

Unfortunately, there are no observations of carbon particles pathways from the surface to the deep sediment traps that could be used for training. To our knowledge, the only method to obtain the source origin of sinking particles captured by moored instruments is a Lagrangian tracking with numerical simulations. The model-based data used for training purposes can exploit simulation datasets as well as reanalysis datasets (e.g: GLORYS). The latter better represents the true ocean state, as mentioned in the introduction. However, they do not fully represent mesoscale / submesoscale and are not validated at depth, which may strongly impact Lagrangian dynamics.

As a result, our strategy was to use a non-assimilated high resolution numerical simulation to represent the ocean dynamic as realistically as possible, with a fine representation of the mesoscale (2 km horizontal resolution < Rossby radius in the region). Although this method

introduces a risk for the network to learn biases associated with the CROCO simulation, we believe that this limitation is likely to be less important compared with the ones mentioned in the discussion. Following an approach similar to Febvre et al. 2024 (https://doi.org/10.1029/2023MS003959) for the neural mapping of altimetry data, future work could explore the impact of the datasets considered for training purposes (e.g., simulation and reanalysis datasets, simulation datasets involving different processes and space-time resolutions) onto the performance of neural networks trained for the prediction of ocean carbon pathways.

2. **Uncertainty in Lagrangian experiments:** The catchment areas used for training and evaluation are themselves derived from Lagrangian backtracking in a numerical model, which, as stated by the authors, are sensitive to mesoscale dynamics reconstruction. These uncertainties inherently affect the reference data used in the learning process. It would strengthen the manuscript to discuss and quantify how these uncertainties may propagate through the ML training and prediction steps.

In order to ensure the robustness of the lagrangian backwarding method, sensitive tests were conducted in Wang et al. 2022 and Picard et al. 2024. In particular, we evaluated the influence of the final PDF depending on the size of the released patch and the number of particles considered. No significant change was observed. However, these observations are directly limited by the simulation resolution (2 km here). Indeed, a finer resolution may lead to different results by enhancing chaotic situations and divergent flows above the trap.

Presently, evaluating the associated uncertainties with a high degree of precision is challenging, given the lack of available numerical simulations at finer scales in the region, which would facilitate such an evaluation. However, one potential solution that avoids the use of an additional simulation is to adopt a stochastic approach (https://doi.org/10.1175/1520-0485(2002)032<0797:MTIOGP>2.0.CO;2), whereby random noise is introduced during the particles' trajectories to account for the absent diffusion processes. The random noise can be, for instance, parameterized based on the local dynamics. Unfortunately this stochastic Lagrangian modelling is not yet incorporated in our Lagrangian tool (i.e, Pyticles ;https://doi.org/10.5281/zenodo.4973786).

Nevertheless, in order to provide an example of the aforementioned method and to illustrate the associated uncertainties, we implemented a simplified Markov model at order 0 in Pyticles. The trajectory $x_n$ of the particles can thus be described as follows:

$$x_{n+1} = x_n + \Delta t\, u(x_n, t_n) + \mathcal{R}(2K\Delta t)^{1/2}$$

The final term is associated with the stochastic implementation, whereby $\mathcal{R}$=N(0,1) denotes a random number selected according to a normal distribution and K represents the diffusivity associated with subgrid scale turbulence. For this example, we chose two constant diffusivities $K = 0.1\, m^2 s^{-1}$ and $K = 1\, m^2 s^{-1}$. Considering a $\Delta t = 120s$, this leads roughly to typical horizontal speed of 0.04 $m\,.\,s^{-1}$ and 0.13 $m\,.\,s^{-1}$.

Two distinct periods are considered in the following: In the first example (Figure 1), the majority of particles are contained within an eddy. In the second example (Figure 2), the flow is more chaotic and the particle source area is more spread out. For each example, the following plots are presented: the original particle PDF without K diffusion, and the Gaussian filtered PDF that was utilised in this study. For each Kdiff, we have conducted ten experiments for each period, and four of them have been plotted. The resulting PDFs are then averaged, and a comparison is drawn between the averaged PDF and the Gaussian-filtered PDF.

[Figure]

Figure 1: Eddy case. First row : 4 examples of PDF with Kdiff = 0.1. Second row respectively from left to right : PDF with Kdiff = 0, PDF Kdiff = 0 after gaussian filter, average of 1 PDF with Kdiff = 1, absolute error between the two previous PDF. Last two rows : Same with

Kdiff= 1.

[Figure]

Figure 2: Chaotic case. First row : 4 examples of PDF with Kdiff = 0.1. Second row respectively from left to right : PDF with Kdiff = 0, PDF Kdiff = 0 after gaussian filter, average of 10 PDF with Kdiff = 1, absolute error between the two previous PDF. Last two rows : same plots with Kdiff = 1.

As shown in the two examples, when Kdiff is set to 0.1, the area of the averaged PDF appears to be comparable to the gaussian-filtered PDF in both cases. However, when Kdiff is set to 1, the averaged PDF is distributed over a larger area and may reveal new potential source regions, especially for chaotic situations (see the new particle patch in the middle left of Figure 1 and around the mesoscale eddy in the upper right of Figure 2). Consequently, a precise parameterization of Kdiff is likely to be of significance in this study and in future

research. It would be worthwhile to further investigate this approach with a more complex diffusion model that parametrizes Kdiff with the local dynamics.

We have therefore added an Appendix D to present the second case and the following point has been added at the end of the discussion:

"*Finally, questions remain about the uncertainties associated with the Lagrangian method. It is clear that the uncertainty associated with the Lagrangian method has a direct impact on the predictions, since the network is trained directly with the backtracked particles. Although sensitivity tests have been carried out (changing the number of particles and the size of the released patch) to ensure that the particle sources are not affected, some diffusion processes are not represented in the numerical simulation and consequently in the propagation of the particles. To evaluate potential biases, it would be necessary in the future to adopt a stochastic approach \citep{Minguez2012}, where random noise is introduced into the particle trajectories to account for subgrid scale diffusion processes. These processes have the potential to influence the results of the Lagrangian analysis (see Appendix D for an example). Consequently, the diffusion parametrization should be carefully defined, taking into account local dynamics. This approach would facilitate the establishment of a confidence interval for the source areas.*"

3. **Sinking velocity and catchment area reconstruction:** The choice of a reference sinking velocity of 100 m d⁻¹, instead of the 50 m d⁻¹ value used in Picard et al. (2024), makes direct comparison between the two studies more difficult (for example it is hard to take the conclusions stated in Section 5.3, line ~400). In the introduction it would be beneficial to explain why it is important to train the model using different particle sinking velocities (summary of the discussion/conclusion in Picard et al. 2024). Moreover, the particle sinking velocity is assumed constant over depth (0–3000 m), the authors could discuss what implications this may have on the reconstruction of catchment areas. Seasonal changes in particle sinking velocities could have been considered in this study as it was mentionend in Picard et al. (2024) and at least it should be more discussed in this paper.

In Picard 2024, the focus was on a simplified case with a 1000 m deep sediment trap and a configuration that included slow-sinking particles to ensure the potential for application at higher particle sinking rates. Initially, it was hypothesised that the consideration of a deeper sediment trap would not significant alter the results due to the weak dynamics below 1000 m. However, this hypothesis was refuted in this study. Ideally, sinking particles of 50 m.d-1 would also be considered in this study to allow a direct comparison. However, this would raise significant technical challenges, namely the necessity to consider a larger domain > 1000 km for the source and a minimum timel window of 2 months for each of the Lagrangian experiments. Nevertheless, considering Figure 4 and the result with a sinking rate of 80 m.d-1, it is reasonable to assume that the majority of predictions at this rate will not be valid.

We added in the introduction an explanation to justify the importance of considering several sinking rates: "*Indeed, as previously stated by Wekerle, 2018, the provenance of particles can vary considerably depending on their sinking velocity. Consequently, it*

*is imperative to consider the entire particle velocity spectrum in order to accurately represent all the possible source areas.*"

We added a discussion concerning the limitation of the Lagrangian experiments :

"*A notable constraint of the study lies in the simplified representation of the organic particles within the numerical simulation. As mentioned in Picard et al, 2024, the size of the particles, and their sinking rate vary during their descent through the water column, due to aggregation/ disaggregation, grazing, remineralization by bacterial activity (Alldredge et al, 1988,Berelson et al, 2001,Fischer et al, 2009,VillaAlfageme et al, 2016). These processes have not been taken into account in the presented Lagrangian experiments, and it is clear that they must be considered in future experiments. One potential approach to achieve this would be to use a Lagrangian framework that incorporates the parameters of particle-biological interactions, as suggested by Jokulsdottir et al, 2016.*"

In addition, it would be interesting to consider seasonal changes in the particle sinking rate, but we believe that today there is a lack of observations that do not allow a clear quantification of the seasonal sinking rate in the region.

4. **Figure 2 :** In panel (c), two different black lines representing "true" PDFs are shown. However, one seems consistent and the other not. Please clarify this in the figure caption and text — currently this is confusing.

In Figure 2 (c) the true PDF is split into two patches, and we can only assume that during the 10 days of release, the particles take different pathways over this period. This often happens when the flow close to the source area is divergent: a small perturbation can lead to a very different particle source area. This seems to be the case here as the source location is between 3 eddy structures. We can therefore assume that the Lagrangian pathway is very chaotic, leading to a challenging prediction, which explains the bad score. However, we do not feel able to draw any conclusions on the fact that one patch is better predicted than another.

Clarification in the caption or above : "*Be advised that in scenario c), the true PDF is split into two patches. This is likely due to divergent dynamics at the source point located at the junction of several eddies, which seems to make the prediction more challenging.*"

5. **Line 175:** The sentence needs to be reformulated for clarity — it is not easy to understand that Unet 5V-1L is used to assess the resolution of SST and SSH data used in Unet sst-ssh. Line 177: "w" should be replaced by "100" in "Unet w 5V-1L". Figure 3 caption: If I well understood, you should replace "Unet 100 SSH-SST" with "Unet 100 5V-1L" as it is this last that helps evaluate Unet 100 SSH-SST.

Indeed the sentence was very confusing. The Unet_ssh-sst is used here for the diagnostic and only SSH and SST is considered as input. We reformulated the sentence :

"To evaluate the robustness of the predictions with respect to the horizontal resolution of the input variables, *we examine the evolution of the prediction score given by Unet_sst-ssh^{100}  (Figure 3) by progressively degrading the effective resolution of the input SST (black dashed line) and SSH (red dashed line) fields from 8 km (effective resolution of the numerical simulation) to 200 km.*"

We also replaced "w" with "100".

6. **Section 3.2 and Figure 4:** The addition of catchment areas derived from fixed-radius boxes (100 km, 200 km) is not clearly tied into the conclusions in the paper. It would be useful to explain what this comparison adds, and to which model configurations these comparisons refer.

In the conclusion we mentioned the boxes references:

 "*The results of our method are compared with the direct use of a 100-200 km box around the trap location,  which represents the conventional approach of catchment area. The results show that the prediction score increases with w, and while the 100-200 km boxes only predict 20-30% of the particles catchment area (w=80-300 m.d^{-1}), the Unet_{sst-ssh} predictions enhance this score to 40-60%.*"

In the discussion at the beginning of the part "5.1 Comparison with previous studies", we also added :

*"Conventionally, this dynamic effect has been addressed by considering a zone of influence, typically represented by a 100 or 200 km box surrounding the sediment trap (Lampitt et al., 2023). However, this approach is limited in its ability to handle the spatial and temporal variability of the source area, which can vary on a weekly basis due to local mesoscale dynamics. The objective of this study is to examine the potential of using surface ocean mesoscale dynamics data to establish a more robust relationship between the deep ocean carbon fluxes from the PAP-SO ST and surface ocean dynamics. This approach has the capacity to make effective predictions using only surface data, thereby improving the source area location compared to a simple box."*

7. **Figure 7 :** It would be useful to include the Unet D100 simulation predictions in Figure 7 to compare with satellite-based results. Moreover, including the true catchment area from the Lagrangian simulation (used for training) would provide valuable context. Also, the black arrows on the map are barely visible — if kept, their size should be increased significantly.

We have changed the size of the arrows in Figure 7 and the other figures to make them easier to see.

The Unet D100 simulation provides predictions from the non-assimilated simulation. The dynamics in this numerical simulation are not directly comparable with the real ocean dynamics at the same period. Therefore, we cannot directly compare the Lagrangian simulation / Unet D100 with the satellite-based results. However, in the future it may be interesting to compare our results with Lagrangian experiments using assimilated data such as GLORYS12V1 (https://doi.org/10.48670/moi-00021).

**In the discussion (line 367) :** the role of phytoplankton community composition is mentioned. The authors could suggest the use of OC-CCI micro/nano/pico phytoplankton data (Copernicus Marine Service, daily, 4 km resolution) as a way forward. This product is operationnally available and could help better assess the community composition impact on fluxes.

We added a sentence in the discussion part :

"In the future, it would be necessary to refine our analysis by taking into account the local plankton communities observed in the catchment areas to include further information such as the sinking rate and the export ratio. ***For instance, the use of OC-CCI micro/nano/pico phytoplankton data (Copernicus Marine Service, daily, 4 km resolution) could facilitate a more comprehensive assessment of the impact of community composition on fluxes, thereby improving our interpretation of the ST data.***"

8. **Biogeochemical-Argo and vertical distribution discussion:**

   ○ Line 376: Please include full name and citation of BGC-Argo (BioGeoChemical-Argo; Claustre et al., 2020).

   Done

   ○ Line 388: Replace citation "Sauzède et al., 2017" with more appropriate reference: Sauzède et al., 2016, JGR Oceans, https://doi.org/10.1002/2015JC011408.

   Done

   ○ Additionally, it could be noted that this method has led to a Copernicus Marine Service product (https://data.marine.copernicus.eu/product/MULTIOBS_GLO_BIO_BGC_3D_ REP_015_010/description) which provides 3D vertical fields of bbp, POC and Chl, and could support future analysis of phytoplankton vertical distribution.

   We added in the discussion :

   "***Some of these products (i.e, 3D fields of Particulate Organic Carbon, Particulate Backscattering coefficient and Chlorophyll-a concentration) are already available (https://doi.org/10.48670/moi-00046).***"

9. **Section 5.3:** Around line 400, it is difficult to draw solid conclusions since both sinking velocity and sediment trap depth are varied between experiments. The authors should clarify this limitation and explain how they can conclude.

We provided some details in the discussion:

"*However, we can hypothesise that the use of a comparable sinking speed of 50 m.d-1 in this study would result in less than 50% of valid predictio*n *(considering that the score decreases with lower w and that at the slowest sinking rate of 80 m.d-1 we only reached \sim 50% of valid predictions with $Unet_{5V-1L}$). Hence,*  this study seems to indicate that the vertical distance from the upper ocean and the resulting influence of local deep dynamics may be more important than initially hypothesised."

10. **Conclusion :** The use of bullet points in the conclusion could be avoided. Writing in full sentences would enhance the flow and readability of the text.

The conclusion has been reformulated as follow :

"*This study presents a novel machine learning tool, named $Unet_{sst-ssh}$, which is capable of predicting the catchment area of particles trapped at the PAP-SO station ST moored at 3000 m depth, based solely on remote sensing data, namely SST and SSH. The study considers five sinking velocities, ranging from 80 m d$^{-1}$ to 300 m d$^{-1}$. The results of our method are compared with the direct use of a 100-200 km box around the trap location,  representing the conventional approach of catchment area. The results show that the prediction score increases with $w$, and while the 100-200 km boxes predict only 20-30\% of the particles catchment area (w=80-300 m.d$^{-1}$), the $Unet_{sst-ssh}$ predictions enhance this score to 40-60\%.*

*We applied $Unet_{sst-ssh}$ to real satellite observations at PAP-SO, resulting in the generation of a 20-year catchment area dataset available at a 10-day resolution. The dataset demonstrated a stronger correlation, and therefore connection, between the deep ocean particle fluxes measured at PAP-SO and surface chlorophyll-a concentration compared to the traditional catchment area method. The presence of deep ocean energetic dynamics that are uncorrelated with the surface appears to be the main reason for the invalid predictions. Future improvements to the $Unet_{sst-ssh}$ method would entail a more comprehensive consideration of these deep currents. Ultimately, the improved identification of the surface catchment area of particles collected in deep ocean sediment traps would facilitate the identification of the surface drivers of deep ocean carbon sequestration, thereby improving our understanding of the biological carbon pump.*

*"*

**Minor Comments and Corrections:**

- **Line 28:** "euphotic zone (~0–200 m)" – consider defining this explicitly and add the ~. Done

- **Throughout:** Replace all instances of "chlorophyll" with "chlorophyll-a" for consistency and accuracy. Done

- **Line 165:** Clarify that true particle origins are derived from Lagrangian experiments; use "PDF" instead of "pdf". Done

- **Figure 2:** Title should be: "Examples of predictions". Done

- **Line 189:** Figure 4. Done

- **Line 192:** This is probably because Done

- **Line 191 and 196:** Add figure or table references to support statements (reference to Figure 4 ?) Done

- **Figure 4:** Label sub-panels with a), b), c); fix caption spacing: "been computed". Done

- **Figure 5 caption:** Add space: "…is shown". Done

- **Line 250-255:** Add links to data Done

- **Line 265 :** specify that satellite Chl-a is from OC-CCI, with appropriate citation. Done

---

## Author Comment (AC2)

We thank the reviewers for their positive and constructive comments. We have addressed all comments as detailed below.

@#R1

The manuscript by Picard et al. evaluates the catchment area of the ST moored 3000 meters below the seafloor. The authors employ machine learning technology to predict this catchment area based on the input of Sea Surface Height (SSH) and Sea Surface Temperature (SST). This study offers a valuable tool for the observational community, and the methodology and results appear convincing. Therefore, I recommend a minor revision with the following suggestions:

1. The authors should address the uncertainty associated with the backward tracing technique. Specifically, how is the confined interval of the backward tracing method evaluated? Since the CNN method is trained using the backward tracing results, a discussion on the propagation of uncertainty and its potential impact on the predictions would enhance the robustness of the approach.

In order to ensure the robustness of the Lagrangian backwarding method, sensitive tests were conducted in Wang et al. 2022 and Picard et al. 2024. In particular, we evaluated the influence of the final PDF depending on the size of the released patch and the number of particles considered. No significant change was observed. However, these observations are directly limited by the simulation resolution (2 km here). Indeed, a finer resolution may lead to different results by enhancing chaotic situations and divergent flows above the trap.

Presently, evaluating the associated uncertainties with a high degree of precision is challenging, given the lack of available numerical simulations at finer scales in the region, which would facilitate such an evaluation. However, one potential solution that avoids the use of an additional simulation is to adopt a stochastic approach (https://doi.org/10.1175/1520-0485(2002)032<0797:MTIOGP>2.0.CO;2), whereby random noise is introduced during the particles' trajectories to account for the absent diffusion processes. The random noise can be, for instance, parameterized based on the local dynamics. Unfortunately this stochastic Lagrangian modelling is not yet incorporated in our Lagrangian tool (i.e, Pyticles ;https://doi.org/10.5281/zenodo.4973786).

Nevertheless, in order to provide an example of the aforementioned method and to illustrate the associated uncertainties, we implemented a simplified Markov model at order 0 in Pyticles. The trajectory $x_n$ of the particles can thus be described as follows:

$$x_{n+1} = x_n + \Delta t\, u(x_n, t_n) + \mathcal{R}(2K\Delta t)^{1/2}$$

The final term is associated with the stochastic implementation, whereby $\mathcal{R}$=N(0,1) denotes a random number sampled according to a normal distribution and K represents the diffusivity associated with subgrid scale turbulence. For this example, we chose two constant diffusivities $K = 0.1\, m^2\, s^{-1}$ and $K = 1\, m^2\, s^{-1}$. Considering a $\Delta t = 120s$, this leads roughly to typical horizontal speed of 0.04 $m\,.\,s^{-1}$ and 0.13 $m\,.\,s^{-1}$.

Two distinct periods are considered in the following: In the first example (Figure 1), the majority of particles are contained within an eddy. In the second example (Figure 2), the flow is more chaotic and the particle source area is more spread out. For each example, the following plots are presented: the original particle PDF without K diffusion, and the Gaussian filtered PDF that was utilised in this study. For each Kdiff, we have conducted ten experiments for each period, and four of them have been plotted. The resulting PDFs are then averaged, and a comparison is drawn between the averaged PDF and the Gaussian-filtered PDF.

[Figure]

Figure 1: Eddy case. First row : 4 examples of PDF with Kdiff = 0.1. Second row respectively from left to right : PDF with Kdiff = 0, PDF Kdiff = 0 after gaussian filter, average of 1 PDF with Kdiff = 1, absolute error between the two previous PDF. Last two rows : Same with

Kdiff= 1.

[Figure]

Figure 2: Chaotic case. First row : 4 examples of PDF with Kdiff = 0.1. Second row respectively from left to right : PDF with Kdiff = 0, PDF Kdiff = 0 after gaussian filter, average of 10 PDF with Kdiff = 1, absolute error between the two previous PDF. Last two rows : same plots with Kdiff = 1.

As shown in the two examples, when Kdiff is set to 0.1, the area of the averaged PDF appears to be comparable to the gaussian-filtered PDF in both cases. However, when Kdiff is set to 1, the averaged PDF is distributed over a larger area and may reveal new potential source regions, especially for chaotic situations (see the new particle patch in the middle left of Figure 1 and around the mesoscale eddy in the upper right of Figure 2). Consequently, a precise parameterization of Kdiff is likely to be of significance in this study and in future

research. It would be worthwhile to further investigate this approach with a more complex diffusion model that parametrizes Kdiff with the local dynamics.

We have therefore added an Appendix D to present the second case and the following point has been added at the end of the discussion:

"***Finally, questions remain about the uncertainties associated with the Lagrangian method. It is clear that the uncertainty associated with the Lagrangian method has a direct impact on the predictions, since the network is trained directly with the backtracked particles. Although sensitivity tests have been carried out (changing the number of particles and the size of the released patch) to ensure that the particle sources are not affected, some diffusion processes are not represented in the numerical simulation and consequently in the propagation of the particles. To evaluate potential biases, it would be necessary in the future to adopt a stochastic approach \citep{Minguez2012}, where random noise is introduced into the particle trajectories to account for subgrid scale diffusion processes. These processes have the potential to influence the results of the Lagrangian analysis (see Appendix D for an example). Consequently, the diffusion parametrization should be carefully defined, taking into account local dynamics. This approach would facilitate the establishment of a confidence interval for the source areas.***"

2.   I suggest discussing the regional dependence of the Unetsst−ssh method in greater detail. In particular, it is my understanding that surface data-based training may be more applicable in regions dominated by geostrophic or quasi-geostrophic currents. In areas with strong submesoscale processes, the bias of the model may increase, and it would be valuable to address this limitation in the context of the study.

Given that SSH is the main driver of the network score (figure 3), we can indeed reasonably assume that the network relies on geostrophic currents for its predictions. As a result, we believe that regions dominated by geostrophic or quasi-geostrophic currents should perform better with this network than, for example, regions with strong submesoscale processes, and this should be evaluated in the future.

We have modified the discussion at the end as follows:

"It is also possible to consider the deployment of sediment traps in a region where the deep dynamics are even weaker and unaffected by the near-topography, which is typically the source of deep eddies and instabilities (Smilenova et al., 2020). A numerical simulation such as the one used here can be used to identify the weakest dynamical regions, where particle pathways below the mesopelagic layer are unlikely to be affected. Nevertheless, 3D numerical simulation remains one of the most effective methods for studying deep ocean dynamics and further efforts are required to validate the accuracy of simulations of deep ocean currents. ***In addition, given that SSH is the main driver of the network score (see Figure 3), we hypothesise that the network relies predominantly on geostrophic currents to perform its prediction. Consequently, it would be worthwhile to compare the efficiency of the model in regions with varying degrees of geostrophic current dominance.***"

3. The manuscript refers to the POF index, but the definition of this index is not provided in the main text. I recommend including a clear definition of the POF index to ensure that readers unfamiliar with the term can follow the methodology and results effectively.

As far as we know, there is no mention of the words "POF" or "index" in the manuscript.

---

## Author Response (AR1)

Dear editor

I am writing to inform you that the manuscript has been reviewed and is now available for your consideration. We believe we have addressed all the concerns raised during the review process and you can find our answers in the interactive discussion.

We appreciate the time and effort that the reviewers and editorial team have dedicated to evaluating our work. Please let us know if there are any further steps we need to take or additional information required from our end.

Thank you for your attention to this matter. We look forward to your positive response.

Sincerely,

Théo Picard